# Estimating raindrop size distributions using microwave link measurements: potential and limitations

Thomas C. van Leth[1], Hidde Leijnse[2], Aart Overeem[1,2], and Remko Uijlenhoet[1]

[1]Hydrology and Quantitative Water Management Group, Wageningen University
[2]R&D Observations and Data Technology, Royal Netherlands Meteorological Institute (KNMI)

**Correspondence:** Remko Uijlenhoet (remko.uijlenhoet@wur.nl)

**Abstract.** We present a novel method of using two or three collocated microwave link instruments to estimate the three parameters of a gamma raindrop size distribution (DSD) model. This allows path-average DSD measurements over a path length of several kilometers as opposed to the point measurements of conventional disdrometers. Our method is validated in a round-trip manner using simulated DSD fields as well as five laser disdrometers installed along a path. Different potential link combinations of frequency and polarization are investigated. We also present preliminary results from the application of this method to an experimental setup of collocated microwave links measuring at 26 GHz and 38 GHz along a 2.2 km path. Simulations show that a DSD retrieval on the basis of microwave links can be accurate under idealized conditions. We found that a triple link retrieval provides little added benefit over a dual-link retrieval in terms of accuracy or precision. In practice, the accuracy and success rate of any retrieval is highly dependent on the stability of the base power level as well as the precision of the instruments and in particular the quantization applied to the recorded power level.

## 1 Introduction

The use of microwave links to measure rainfall intensity has received significant attention in the last decade. The main driver for this has been the insight that the backhaul links of mobile communication networks are suitable for such measurements and are available in greater numbers than dedicated rainfall measurement stations in many countries (Messer et al., 2006; Leijnse et al., 2007b; Gosset et al., 2016). The usage of microwave uplinks to geostationary satellites has been investigated as well (Giannetti et al., 2017). In addition, efforts have been taken to expand the range of atmospheric phenomena that can be measured with microwave links, including fog (David et al., 2013), solid precipitation (Cherkassky et al., 2014) and evapotranspiration (Leijnse et al., 2007a). Another enticing possibility is the use of multiple link instruments along the same path to measure not just the bulk rainfall intensity, but the path-average raindrop size distribution (DSD).

DSD estimates can be used to derive all other bulk rainfall variables and are therefore valuable for a variety of purposes (see e.g. Uijlenhoet and Stricker (1999) for an overview of relevant statistical moments). Examples include precipitation microphysics (e.g. Uijlenhoet et al. (2003)), soil erosion by rain (e.g. Angulo-Martínez and Barros (2015); Salles et al. (1999); Salles and Poesen (1999, 2000)) and radar validation (e.g. Hazenberg et al. (2014)). The use of microwave links for such purposes

promises measurements that are more spatially representative of radar or satellite pixels than what is offered by the usually sparse networks of impact, laser and video disdrometers that are currently the most common way DSDs are measured.

In order to estimate the DSD from a limited number of statistical moments, a parameterization has to be used. The gamma distribution with three parameters provides a good estimate for a wide variety of rainfall types (Ulbrich, 1983). Rincon and Lang (2002) were first to attempt a gamma DSD parameter retrieval using two microwave links, with promising results. However, their methods require an a priori parameter estimate in addition to the parameters derived from the two measurements. To the best of our knowledge, no further research has been published regarding DSD estimations using microwave links. However, the adjacent field of polarimetric radar measurement has seen much development (see e.g. Fabry (2015)) and many different methods to estimate DSDs from polarimetric radar have now been developed and tested.

A handful of different techniques can be identified that have been developed to retrieve DSD parameters from a limited number of polarimetric radar moments: The constrained gamma method developed by Zhang et al. (2001), the $\beta$ method proposed by Gorgucci et al. (2002) and double-moment DSD normalization (Raupach and Berne, 2017). Several machine learning approaches have also been used: neural networks (Vulpiani et al., 2006), Bayesian regression (Cao et al., 2010) and tree-based genetic programming (Islam et al., 2012).

In this paper we will explore the potential of a numerical retrieval from microwave attenuation and/or differential propagation phase shift. We consider two different techniques: The first method uses three measured microwave link variables to derive the three parameters of the gamma distribution. Here, the gamma parameters are weakly constrained (i.e., only a limited range of parameter values is allowed). The other method uses two measured microwave link variables to derive two parameters of the gamma distribution. Here, the third parameter is completely constrained by the other two, similar to the method used by Zhang et al. (2001) for radar moments. We will apply these techniques first using a simulated dataset based on radar and disdrometer measurements from the Ardèche region, France. Next, we will apply the method to microwave link variables derived from a set of path-averaged measurements from five laser disdrometers in Wageningen, The Netherlands. Finally, we will test the methods on measurements from microwave link instruments along the same path as the disdrometers in Wageningen (Van Leth et al., 2018) and compare the resulting DSDs with the DSDs measured from the disdrometers.

The paper is organized as follows: In Section 2 we present in more detail the datasets used in this paper. In Section 3 we present the theory and the methods used to retrieve DSDs from microwave link variables. We will also describe the validation methods employed. In Section 4 we discuss the results retrieved from the Ardèche dataset. In Section 5 we discuss the results of several tests using simulated attenuations from the Wageningen disdrometer dataset. We also consider the efficacy of different frequency combinations and the robustness of the retrieval to measurement uncertainty. In Section 6 we will apply the developed methods to actual link measurements that were obtained in Wageningen. In Section 7 we will present our thoughts on the feasibility of the techniques in practice and the choices made in this paper. Finally, in Section 8 we come to conclusions and give recommendations for further study.

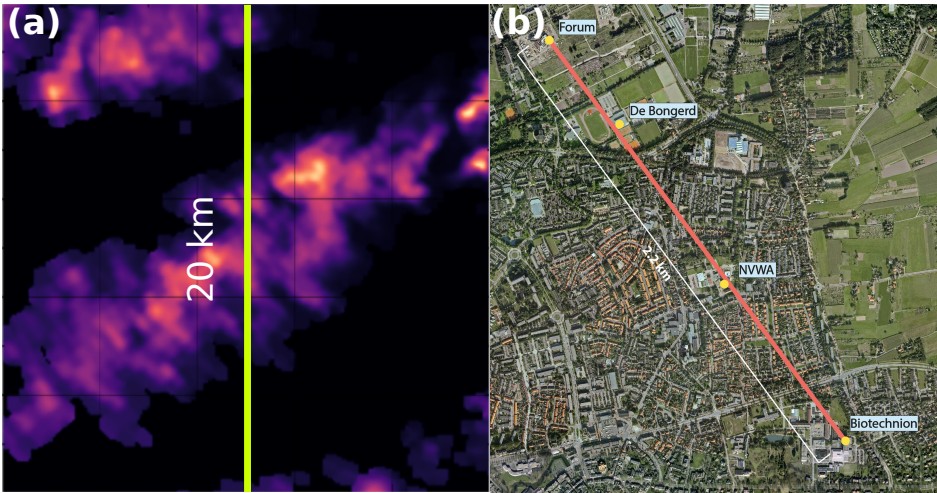

**Figure 1.** a) Sample rain field from the Ardèche dataset with the chosen transect. b) Positions of the disdrometers and the link path in the Wageningen dataset.

## 2 Data

### 2.1 Ardèche DSD reanalyses

Our first test-case consists of a 2-dimensional interpolated DSD field based on polarimetric radar data measured in the Ardèche, France as part of the HyMeX campaign (Raupach and Berne, 2016, 2017). This field was generated using an advection-based temporal interpolation technique (De Vos et al., 2018), where disdrometer data was used to train the technique. This technique does not produce $\mu$ and $\Lambda$ parameters, but instead produces a full DSD histogram per pixel per timestep. The field has spatial dimensions of $20 \times 20$ km and a resolution of 100 m. It covers two distinct events on 27 November 2012 and 27 October 2013. Their durations are in the order of several hours and the timestep is 30 s. Both events can be classified as orographic or convective events. The second event is more spatially heterogeneous than the first, with a decorrelation distance of 2.8 km vs 11 km at 30 s accumulation intervals. The DSD is divided in 20 diameter bins with unequal bin width based on the detection bins of the OTT Parsivel[2] laser disdrometer measurements upon which the reanalysis is partially based. The smallest drop diameter class is 0.3 mm, while the largest diameter is 6.5 mm. A more detailed description of the dataset and the technique used to generate it can be found in De Vos et al. (2018).

We take a transect through this field over the entire length of the field (Fig. 1a) and average the DSD over this transect to approximate the footprint of a microwave link. We calculate the attenuations and differential phase shifts for a number of frequencies: 15 GHz, 26 GHz, 32 GHz and 38 GHz. This includes the frequencies employed in the Wageningen experiment (see below) and covers the range of most commonly employed carrier frequencies in mobile phone networks.

In order to calculate the microwave link variables from the DSD data we first need to interpolate the data in the diameter dimension from the irregular bins to a regular diameter grid with a resolution of 0.1 mm. The main reason for this is to

accurately reproduce the shape of the scattering cross sections as a function of diameter, as simply assuming the scattering cross section to be constant for the entire diameter interval would introduce too much error. We have used a simple linear interpolation method for this purpose.

## 2.2 Wageningen link experiment

Our second test case is a microwave link experimental setup in Wageningen, The Netherlands (Van Leth et al., 2018). The setup consists of three collocated microwave links arranged between two buildings 2.2 km apart and covering mostly built-up terrain. The setup contains one dual-polarization 38 GHz link, which also measures the phase difference between the two polarizations, one additional single polarization 38 GHz link (not used for this paper) and a single polarization 26 GHz link (all with sampling frequencies of 20 Hz; down-sampled, i.e. averaged, to 30 s intervals). In addition, five OTT Parsivel laser disdrometers (providing DSDs at 30 s intervals) are positioned at four locations beneath the link path, including the sites of the transmitting and receiving antennas, as shown in Fig. 1b. The data used in the following analyses is all taken from the period between 1 April 2015 and 1 January 2016. We also specifically focus on one event on 27 July 2015 for illustrative purposes.

In order to use the disdrometer data to represent the DSD of the link path we take a weighted spatial average over the disdrometer data. As with the Ardèche data, we interpolate the DSD data in the diameter dimension from the irregular bins to regular intervals before calculating the microwave link variables. In order to improve the robustness of the results, we only consider measurement intervals where each one of the five disdrometers counted at least 50 drops (see Uijlenhoet et al. (2006) for the effect of drop sample size on the robustness of the DSD estimate). We also apply the correction method of Raupach and Berne (2015) to the disdrometer DSDs.

## 3   Methods

### 3.1   Basic procedure

To determine the underlying DSD from a limited number of statistical moments we need an approximation with a limited number of parameters. One of the most widely used approximations for rain DSD is the gamma distribution suggested by Ulbrich (1983),

$$N(D) = N_0 D^\mu e^{-\Lambda D}, \tag{1}$$

where $D$ is the drop diameter in mm, and $N_0$, $\mu$ and $\Lambda$ are the parameters determining the shape and drop concentration. The parameter $\mu$ is a dimensionless shape parameter and $\Lambda$ is a slope parameter with a unit of $\mathrm{mm}^{-1}$. Note that the dimension of $N_0$ is dependent on the value of $\mu$. Therefore it is convenient to also use a derived parameter,

$$N_T = N_0 \Lambda^{-(\mu+1)} \Gamma(\mu+1), \tag{2}$$

where $\Gamma$ is the gamma function. $N_T$ has the unit $\mathrm{m}^{-3}$ and is equal to the total drop concentration (assuming integration limits of 0 to $\infty$), resulting in

$$N(D) = N_T \frac{\Lambda}{\Gamma(\mu+1)} (\Lambda D)^\mu e^{-\Lambda D}. \tag{3}$$

The specific attenuation of a link signal in $\mathrm{dB\,km}^{-1}$ at a given frequency can be described in terms of the DSD,

$$k_H(\lambda) = C_k \frac{\lambda^2}{\pi} \int_0^\infty \Im[S_{HH}(\lambda,D)]N(D)dD \tag{4}$$

$$k_V(\lambda) = C_k \frac{\lambda^2}{\pi} \int_0^\infty \Im[S_{VV}(\lambda,D)]N(D)dD, \tag{5}$$

where $\lambda$ is the wavelength of the incoming and outgoing waves in $\mathrm{mm}$, $\Im$ is an operator indicating the imaginary part of its argument and $C_k = 10^{-3} \ln(10)^{-1} \mathrm{\,dB\,m^3\,km^{-1}\,mm^{-2}}$ is a unit conversion factor. $S_{HH}$ and $S_{VV}$ (dimensionless) are the diagonal components of the forward scattering amplitude matrix $\mathbf{S}$ defined by the relationship

$$\boldsymbol{E} = \mathbf{S} \cdot \boldsymbol{E_0}, \tag{6}$$

where $\boldsymbol{E_0}$ is the electric field strength of the incoming electromagnetic (EM) wave and $\boldsymbol{E}$ is the electric field strength of the outgoing EM wave. The specific phase shift (in $\mathrm{rad\,km}^{-1}$) between the horizontal and vertical components of an outgoing EM wave of a given frequency due to forward scattering can also be described as an integral over the DSD,

$$\phi(\lambda) = C_\phi \frac{\lambda^2}{\pi} \int_0^\infty \Re[S_{HH}(\lambda,D) - S_{VV}(\lambda,D)]N(D)dD, \tag{7}$$

where $\Re$ is an operator indicating the real parts instead of the imaginary parts of the diagonal components of the scattering amplitude matrix. $C_\phi = 10^{-3} \mathrm{\,m^3\,km^{-1}\,mm^{-2}}$ is another unit conversion factor.

The corresponding dimensionless scattering efficiencies are defined as

$$Q_H(\lambda,D) = \frac{4\lambda^2}{\pi^2 D^2} \Im[S_{HH}(\lambda,D)] \tag{8}$$

$$Q_V(\lambda,D) = \frac{4\lambda^2}{\pi^2 D^2} \Im[S_{VV}(\lambda,D)] \tag{9}$$

$$Q_\phi(\lambda,D) = \frac{4\lambda^2}{\pi^2 D^2} \Re[S_{HH}(\lambda,D) - S_{VV}(\lambda,D)]. \tag{10}$$

It is the subtle differences in the shapes of the scattering efficiencies as functions of diameter for different frequencies and polarizations that contain the information necessary to retrieve the DSD. We calculate the scattering amplitude matrix using

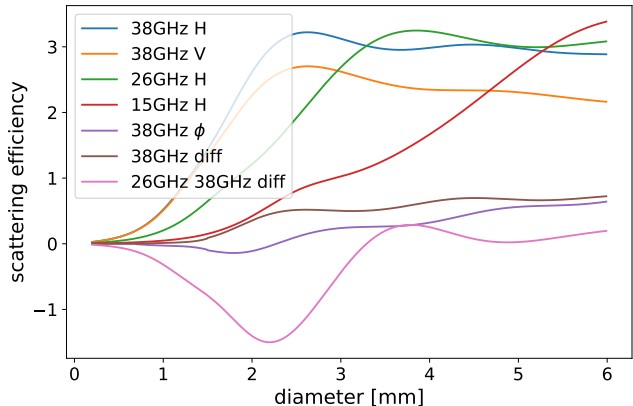

**Figure 2.** Scattering efficiency of raindrops as a function of drop volume-equivalent diameter modeled with the T-matrix method, at a temperature of 288 K.

the T-matrix method (Waterman, 1965) following the approach of Mishchenko et al. (1996); Mishchenko and Travis (1998); Mishchenko (2000). We assume the drops to be oblate spheroids with axis ratios dependent on the drop diameters following the relationship of Thurai et al. (2007) and averaging over canting angles following a Gaussian distribution with a mean of 0° (vertical) and a standard deviation of 2°. Resulting scattering efficiencies at some relevant frequencies are shown in Fig. 2 as a function of volume-equivalent diameter.

The variables used as input in the retrieval could be attenuation of horizontally or vertically polarized radiation or phase differences between horizontally and vertically polarized radiation at one or several frequencies. In order to be able to use attenuations and phase differences interchangeably in the retrieval algorithm we rearrange Eqs (4), (5) and (7) to have the same form:

$$k_H^* = \int_0^\infty \sigma_H(D)N(D)dD \tag{11}$$

$$k_V^* = \int_0^\infty \sigma_V(D)N(D)dD \tag{12}$$

$$k_\phi^* = \int_0^\infty \sigma_\phi(D)N(D)dD, \tag{13}$$

where $k_H^* = k_H/C_k$, $k_V^* = k_V/C_k$, $k_\phi^* = \phi/C_\phi$, and the scattering cross-sections $\sigma_X$ are the scattering efficiencies (Eqs (8)–(10)) multiplied by the drop cross-section ($\frac{1}{4}\pi D^2$): $\sigma_H = \frac{\lambda^2}{\pi}\Im[S_{HH}]$, $\sigma_V = \frac{\lambda^2}{\pi}\Im[S_{VV}]$ and $\sigma_\phi = \frac{\lambda^2}{\pi}\Re[S_{HH} - S_{VV}]$. We will from here on refer to an arbitrary input microwave link variable as $k_i^*$, where $i \in 1, 2, 3$, such that

$$k_i^* = \int_0^\infty \sigma_i(D)N(D)dD. \tag{14}$$

### 3.2 Three-parameter method

Inserting Eq. (1) into Eq. (14) and taking the ratio of two variables we arrive at the following set of equations:

$$\begin{aligned}
\frac{k_1^*}{k_2^*} &= \frac{\int_0^\infty \sigma_1(D)D^\mu e^{-\Lambda D}dD}{\int_0^\infty \sigma_2(D)D^\mu e^{-\Lambda D}dD} \\
\frac{k_2^*}{k_3^*} &= \frac{\int_0^\infty \sigma_2(D)D^\mu e^{-\Lambda D}dD}{\int_0^\infty \sigma_3(D)D^\mu e^{-\Lambda D}dD}.
\end{aligned} \tag{15}$$

Note that these ratios do not depend on the $N_0$ or $N_T$ parameter. If we now replace the integrals by discrete summations we can use an iterative nonlinear root-finding technique to find values for $\mu$ and $\Lambda$ from two ratios of microwave moments. Knowing $\mu$ and $\Lambda$, we can directly solve the discretized equation of one of the microwave link variables for $N_T$.

Fig. 3 shows several possible ratios of microwave link variables as a function of $\mu$ and $\Lambda$. These observables are calculated using the forward model of Eq. (15). Because it is possible to detect the attenuation of the horizontally polarized signal and the vertically polarized signal and the phase difference at a given frequency using a single set of antennae, we prefer this combination of variables over combinations of attenuations at multiple frequencies. We can see in Fig. 3 that the ratios $k_H^*/k_V^*$ and $k_\phi^*/k_H^*$ provide complementary information and are therefore in principle suitable.

We use the Powell hybrid method (Powell, 1970) to solve the system of equations of Eq. (15). We also tested several other gradient-based root-finding methods such as Levenberg-Marquardt and found that this makes little difference in the stability of the retrieval. The convergence of the retrieval is highly dependent on the initial guess. This is problematic, as we want to automatically retrieve a large number of DSDs without manual input. In many retrieval attempts the gradient-based root-finding algorithm diverges towards infinity. In order to prevent this we restrict the root finding algorithm to a limited range of parameter values. If the estimates reach the edge of the range, we reset the parameters to a new initial guess which is taken from within the allowed range, but offset from the first guess by $\Delta\Lambda = 0.2\,\mathrm{mm}^{-1}$ or $\Delta\mu = 0.2$. We systematically traverse the $\mu$-$\Lambda$ space in this way starting from $(-1, 0)$ until convergence is reached.

Even when the algorithm converges, the solution of the system of equations is not necessarily unique. We need an extra set of constraints to make sure we retrieve the parameters that are the most plausible. In order to do so we use the parameters retrieved by the analytical method of moments of Tokay and Short (1996) (TS96) as a rough indication of plausible combinations of $\mu$ and $\Lambda$ values. We have calculated the parameters using this method for all individual timesteps in the 9-month record of the spatially weighted average of the disdrometers placed in Wageningen. Figure 4b shows all these individual retrievals in the

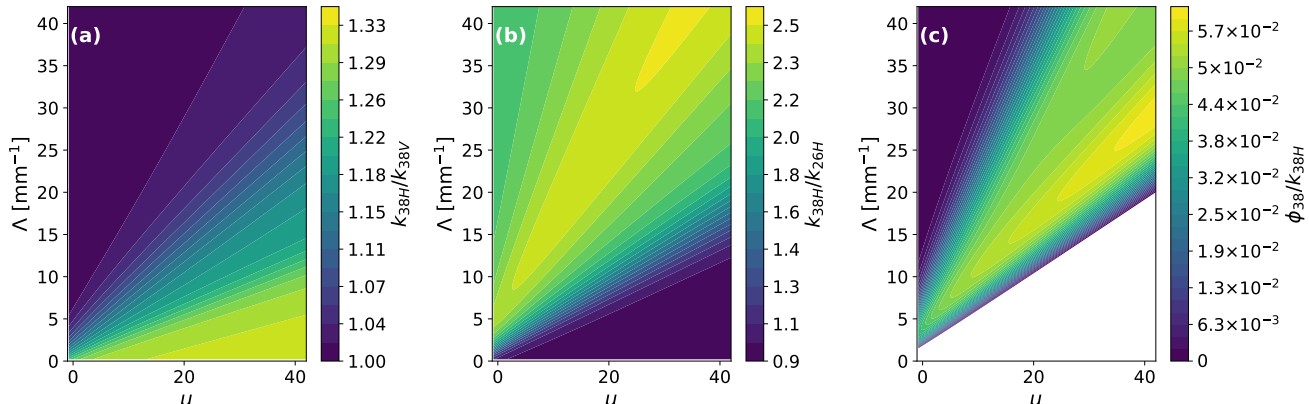

**Figure 3.** Several ratios of microwave link observables that can be used as input to the DSD retrieval, as a function of parameters $\mu$ and $\Lambda$. a) $k_{38H}/k_{38V}$, b) $k_{38H}/k_{26H}$ and c) $\phi_{38}/k_{38H}$.

$(\Lambda, \mu)$ space. We then apply a kernel density estimator to this dataset (shown in Fig. 4a) and calculate the total fraction of data points contained within the contour lines. We then choose the contour corresponding to the 0.95 quantile as a mask. This mask is also shown in Fig. 4b.

The mask is then applied to all attenuation/phase-based numerical retrievals to define the range of allowed parameter values. If the estimate is outside this contour, we reset the root-finding algorithm with a new initial guess that is within the contour, but slightly perturbed from the previous initial guess as described above. This is continued until we find convergence within the contour or we reach the maximum number of iterations without a solution.

### 3.3 Two-parameter method

When only two microwave link variables are available the system of equations of Eq. (15) is reduced to just one equation:

$$\frac{k_1^*}{k_2^*} = \frac{\int_0^\infty \sigma_1(D)D^\mu e^{-\Lambda D}dD}{\int_0^\infty \sigma_2(D)D^\mu e^{-\Lambda D}dD}. \tag{16}$$

In order to still solve for the two parameters an additional equation is required for the relationship between $\mu$ and $\Lambda$. We obtain this relationship empirically. We obtain gamma DSD parameter values via the analytical method of moments from the Wageningen disdrometer dataset. We then fit a second order polynomial function (as shown in Fig. 4b) to the values of $\mu$ and $\Lambda$ with a linear least-squares method. All 9 months of data were used to fit this function. The resulting relationship is

$$\Lambda = 2.5 \cdot 10^{-2}\mu^2 + 1.0\mu + 2.0, \tag{17}$$

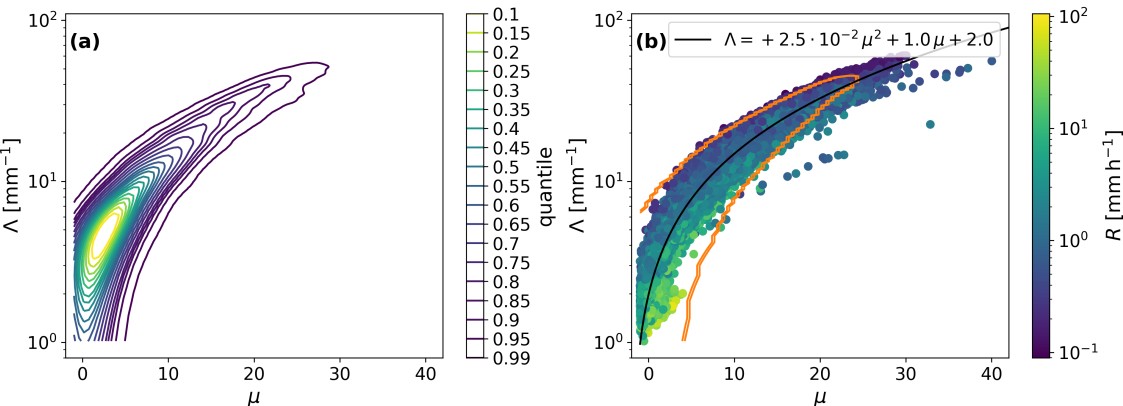

**Figure 4.** a) Density plot of gamma DSD parameters retrieved with the TS96 method, where density contours are given as quantiles enclosed within the contour. Results are for the complete 9-month set of path-averaged disdrometer data. b) Polynomial fit of the $\Lambda$ and $\mu$ parameters in the gamma distribution function fitted to parameter values obtained from the TS96 method of moments, based on the 9 months of path-averaged disdrometer data from Wageningen. The colors indicate the corresponding rain intensities. The 0.95 quantile contour that is used as a masked is overlaid in orange.

which is similar in magnitude but still somewhat different to the relationship found by Zhang et al. (2003). We can substitute this equation for $\Lambda$ in Eq. (16) and then solve for $\mu$ using Brent's root finding method (Brent, 1973). We prefer this method because it is not based on gradients and therefore guaranteed to find a solution if it exists. It is, however, not applicable to multivariate problems.

Figure 5 shows the ratio of Eq. (16) as a function of $\mu$ for two combinations of microwave attenuations that can be measured with the Wageningen link setup: two polarizations at 38 GHz and two frequencies (26 GHz and 38 GHz) at horizontal polarization. From this we can see that the dual-polarization configuration is more suitable for retrieving the DSD; the dual-frequency configuration has non-unique solutions for high underlying values of $\mu$, whereas the dual-polarization model is monotonously decreasing over the entire range of valid $\mu$ values. On the other hand, the dual-frequency ratio of attenuations is much more sensitive to changes in $\mu$ for $\mu$ between $-2$ and $8$, potentially yielding more accurate estimates of this parameter. For the dual-polarization ratio (at 38 GHz) it can also be seen that ratios lower than 1 and higher than 1.25 are not valid. If such ratios are observed they would yield no solution.

### 3.4 Validation methods

We test the capability of the methods to accurately retrieve DSDs and their associated statistical moments with two different datasets of drop size distributions: one simulated and one measured. We use Eqs. (11), (12) and (13) to calculate the microwave link variables from the known DSDs. We then use those variables as input to the retrieval algorithms described in the previous sections to retrieve the parameters of the gamma distribution approximating the DSD. From those parameters the complete

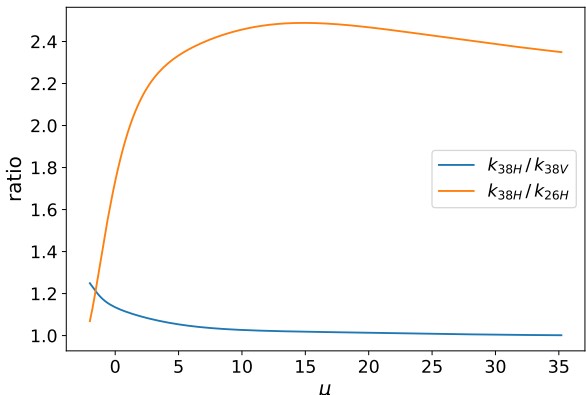

**Figure 5.** Ratios of attenuations as a function of the $\mu$ parameter for the two-parameter retrieval method. the blue lines indicates the attenuation ratio for the dual-polarization method (38 GHz) and the orange line indicates the attenuation ratio of the dual-frequency method (38 GHz and 26 GHz).

DSD and its statistical moments are reconstructed. We then compare these to the original DSD and its moments to assess the systematic bias and random error in the retrieval. To be able to distinguish between cases where the gamma distribution is simply not a good fit for the measured DSD and cases where the retrieval itself is the cause for inaccuracies we also calculate the gamma parameters using the TS96 method. If the gamma distribution is not a good fit for the measured DSD then the results from the TS96 method would deviate appreciably from the measured DSD. If this is not the case, but the retrieved DSD does deviate considerably from the measured DSD, then the retrieval itself is the main cause of the inaccuracy in the DSD. The TS96 method can be used to directly evaluate the parameters of the distribution themselves as well if we assume that the parameters yielded by the TS96 approach are the 'true' parameters.

In order to assess the performance of the retrieval methods we will use a number of statistical measures throughout the results section. As a measure of the accuracy of the retrieval we use the median of the residuals (MOR). As a measure of the precision of the retrieval we use the median absolute deviation of the residuals with respect to the median of the residuals (MAD). We chose MAD and MOR over the use of standard deviation and means, because there are a relatively small number of extreme deviations which would otherwise have too much influence and thus would not give much information about the typical precision. The statistical metrics employed here are less influenced by non-normality and outliers. Because the number and severity of the extreme deviations is also an important part of the performance assessment of the retrieval, we also compute the 95th percentile absolute deviation with respect to the median of the residuals (95AD). Together with the MAD this gives a more complete picture of the distribution of the errors, while still being insensitive to the true outliers. All metrics are normalized with respect to the median of the original quantities, hence they are dimensionless. Furthermore, we also compute the fraction of non-convergent retrievals compared to the total number of retrievals (taking into account the filtering described

in Section 2.2). This 'failure ratio' (also dimensionless) is necessary for a complete picture of the robustness of the method since the other metrics naturally exclude these intervals.

## 4 Validation using simulated DSDs

### 4.1 Single retrieval

Figure 6a shows a typical three-parameter retrieval result from the Ardèche dataset using horizontal attenuation, vertical attenuation and phase difference at a single frequency (38 GHz or 26 GHz in this case). The normalized difference between the retrieved DSD and the original simulation procedure,

$$\Delta N^*(D) = \frac{N_r(D) - N_m(D)}{N_T}, \tag{18}$$

where $N_r$ is the retrieved DSD, $N_m$ is the original DSD and $N_T$ is the integral over all diameters of the original DSD, is 225 illustrated in Fig. 6c. $\Delta N^*$ is within $10^{-4}$ for particles larger than 1 mm. However, at the smallest sizes the results tend to diverge up to $4.5 \cdot 10^{-3}$. In practice we are often more interested in quantities that scale with the statistical moments of the DSD rather than in the DSD itself. Important quantities are e.g. liquid water content $W$ (3rd order moment), rain intensity $R$ (close to 4th order) and radar reflectivity $Z$ (6th order). For higher order moments, the contribution of the smallest drop sizes decreases. To illustrate this we also show the specific rain intensity as a function of drop diameter,

$$230 \quad r(D) = \frac{\partial R(D)}{\partial D}, \tag{19}$$

for the same retrieval parameters in Fig. 6b. The normalized difference in the specific rain intensity (illustrated in Fig. 6d) is given by

$$\Delta r^*(D) = \frac{r_r(D) - r_o(D)}{R_o}, \tag{20}$$

where $R_o$ is the total rain intensity based on the original DSD, $r_r(D)$ is the retrieved diameter-specific rain intensity and 235 $r_o(D)$ is the original diameter-specific rain intensity. From Fig. 6d it can be seen that $-10^{-3} < \Delta r^* < 10^{-3}$ for all drop sizes. The difference in the total drop concentration is $\Delta N_T < 0.2 \cdot N_T$ in the first case, while the difference in the total rain intensity is $\Delta R < 0.03 \cdot R_o$. The assessment of the accuracy of such a retrieval must therefore take into account its most likely application. We will focus our attention in this section on the rainfall intensity and to a lesser extent on the individual gamma DSD parameters. In further sections, when more detailed comparison is required, we will also look at a range of integer 240 moments.

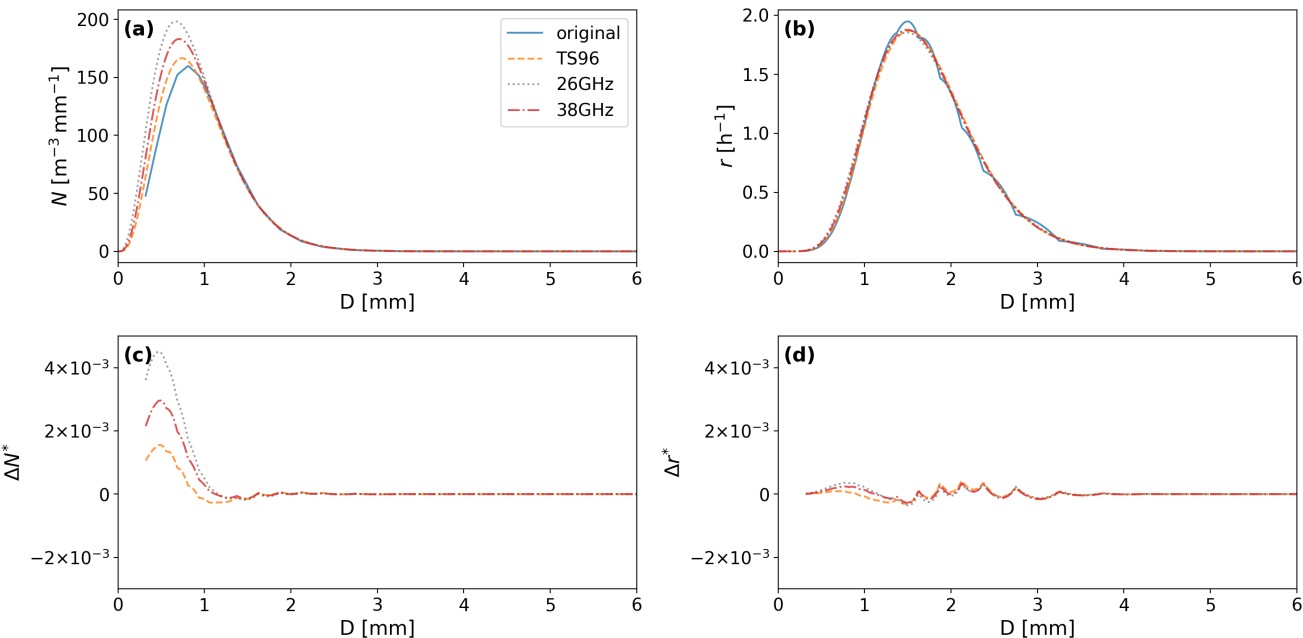

**Figure 6.** a) DSD retrieved from a single timestep in the first Ardèche event using several different microwave frequencies. b) Specific rain intensity for the same timstep. c) Relative difference in DSD compared to the original DSD. d) Relative difference in specific rain intensity compared to original DSD.

## 4.2 Complete events

Figure 7 shows the results of the 3-parameter retrieval for the first Ardèche event, which took place from 26 November 2012 at 4:54 UTC to 27 November at 4:36 UTC. The total duration is almost 24 h and the precipitation intensity averages at 1.77 $\mathrm{mm\,h^{-1}}$ with a maximum of 10.29 $\mathrm{mm\,h^{-1}}$. The gamma DSD parameters of the retrieval are mostly very close to those of the

TS96 procedure, with a few rather large exceptions, particularly noticeable in the $N_T$ parameter. These outliers do not seem to correspond with any particularly high or low precipitation intensity, but they do correspond with high drop concentrations. The temporal evolution of $\mu$ is very close to the temporal evolution of $\Lambda$, with a correlation coefficient of 0.86 (not shown). Similar results can be observed for the second event (see Fig. 8), which took place on 27 October 2013 from 03:22 UTC to 8:48 UTC, with a total duration of 3.5 h, an average intensity of 1.58 $\mathrm{mm\,h^{-1}}$ and a maximum intensity of 27.05 $\mathrm{mm\,h^{-1}}$.

The correlation coefficient between $\mu$ and $\Lambda$ is $\rho = 0.87$. Outliers for this event also occur at lower drop concentrations.

Looking at the rainfall intensity for both events (Fig. 7d and Fig. 8d), we see that the retrieval not only corresponds closely with the TS96 approach, but also with the rainfall intensity derived from the original DSD. There are only a handful of outliers here, the exact timing of which is dependent on the carrier frequencies for which the retrieval is attempted. The MOR, MAD and 95AD of the rain intensity are given in Table 1 for retrievals based on several different carrier frequencies. Another way

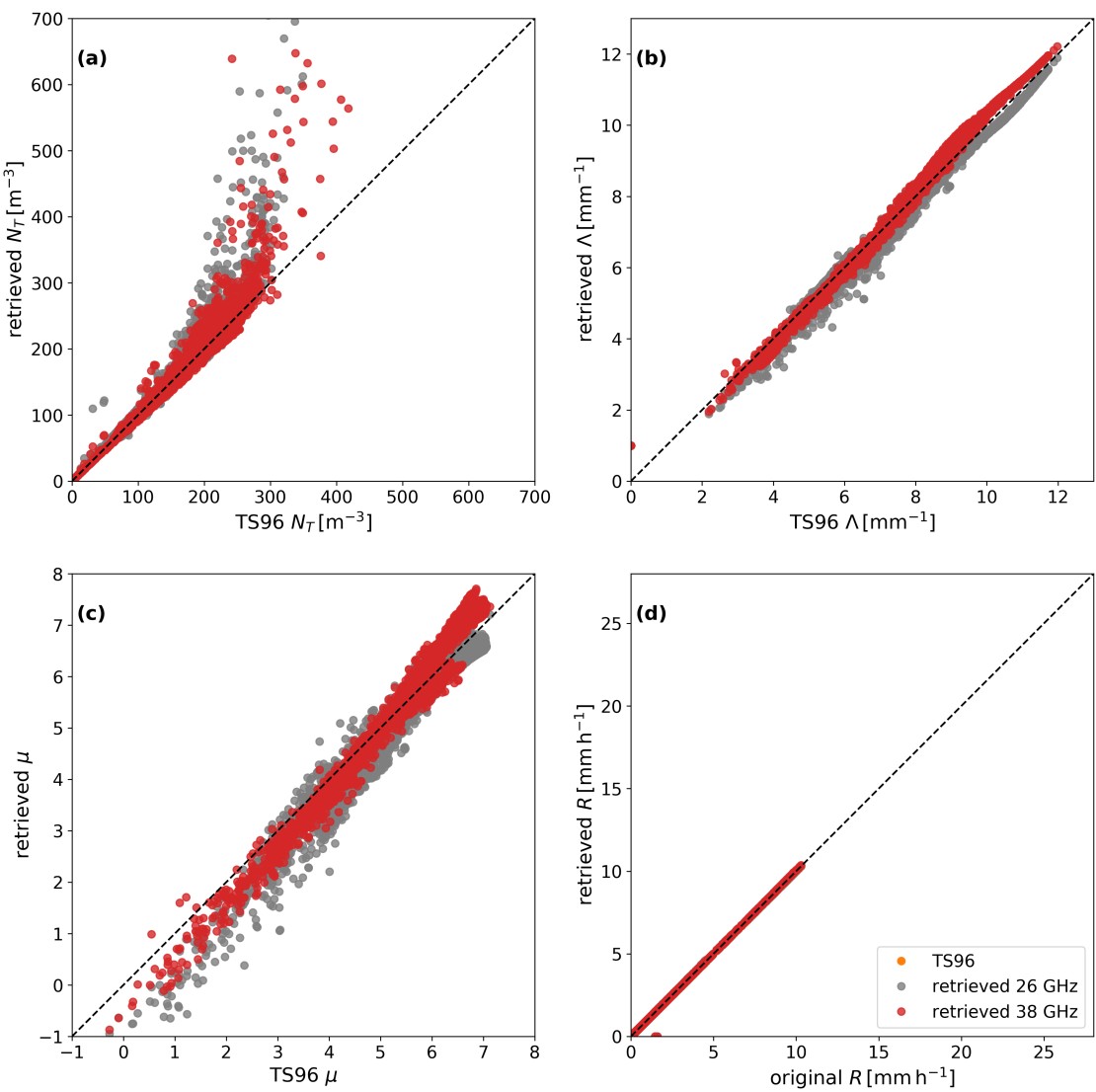

**Figure 7.** Gamma distribution parameters (a–c) and rainfall rates (d) retrieved from a simulation based on an event measured in the Ardèche region on 26/27 November 2012. The retrievals are performed using the three-parameter numerical microwave method and the TS96 method is shown as a reference. The dotted lines indicate a one-on-one relation.

**Table 1.** Statistics of rainfall intensity $R$ relevant to the accuracy and precision of the three-parameter retrieval for two simulated events from the Ardèche region. All statistics are normalized with respect to the median of the original rainfall intensity.

| | 15 GHz | 26 GHz | 32 GHz | 38 GHz |
|---|---|---|---|---|
| **Event 1 (mean: 1.77 mm h$^{-1}$, median: 1.08 mm h$^{-1}$)** | | | | |
| MOR | 0.0048 | 0.0084 | 0.0068 | 0.0049 |
| MAD | 0.0037 | 0.0046 | 0.0042 | 0.0035 |
| 95AD | 0.355 | 0.0202 | 0.0298 | 0.0362 |
| **Event 2 (mean: 1.58 mm h$^{-1}$, median: 0.68 mm h$^{-1}$)** | | | | |
| MOR | 0.0017 | 0.0028 | 0.0045 | 0.0044 |
| MAD | 0.0065 | 0.0040 | 0.0037 | 0.0036 |
| 95AD | 0.2304 | 0.0036 | 0.1197 | 0.0886 |

of assessing the performance of the retrievals is to consider the temporal mean of the DSD. The retrieval (not shown) gives an overestimation for small diameters (with the exception of very low carrier frequencies), while becoming more accurate for larger diameters. For diameters larger than 1 mm the residuals are lower than 0.1 m$^{-3}$ mm$^{-1}$.

## 5 Retrieval from disdrometers

We apply both the two-parameter and three-parameter retrieval algorithms to the Wageningen disdrometer dataset. The results are shown in Fig. 9. We selected a single rain event on 27 July 2015 starting at 9:40 UTC and ending at 12:00 UTC. The first notable difference between the Ardèche and the Wageningen datasets is that in the latter the values for $\mu$ and $\Lambda$ are generally much higher. For both the TS96 approach and the numerical retrieval, values higher than 20 occur regularly for both the $\mu$ and the $\Lambda$ parameter, which is not consistent with what is typically found in the literature (e.g. Raupach and Berne (2016) for the Ardèche, Zhang et al. (2003) for Florida and Atlas and Ulbrich (2006) for Kapingamarangi Atoll (western tropical Pacific)). We can also see that the two-parameter retrieval yields an overestimation of the total drop concentration, while the $\mu$ parameter gets underestimated. Because these biases compensate, this results in a rainfall intensity that is not significantly biased from the original. The three-parameter retrieval method yields a significant overestimation of $\mu$ and $\Lambda$, including a large number of outliers. The three-parameter retrieval produces a less accurate result in terms of rainfall intensity as well. Furthermore, for both retrieval methods, when averaged over an entire event, the retrieval (not shown) overestimates the number of drops with a diameter smaller than 1 mm and underestimates the number of drops with a diameter between 1 and 7.5 mm when compared to the measured DSD (except for very low carrier frequencies). The correlation between the TS96 derived $\mu$ and $\Lambda$ parameters is very high at $\rho = 0.96$ for the two-parameter retrieval. The MOR, MAD and 95AD are given in Table 2. In general, the precision

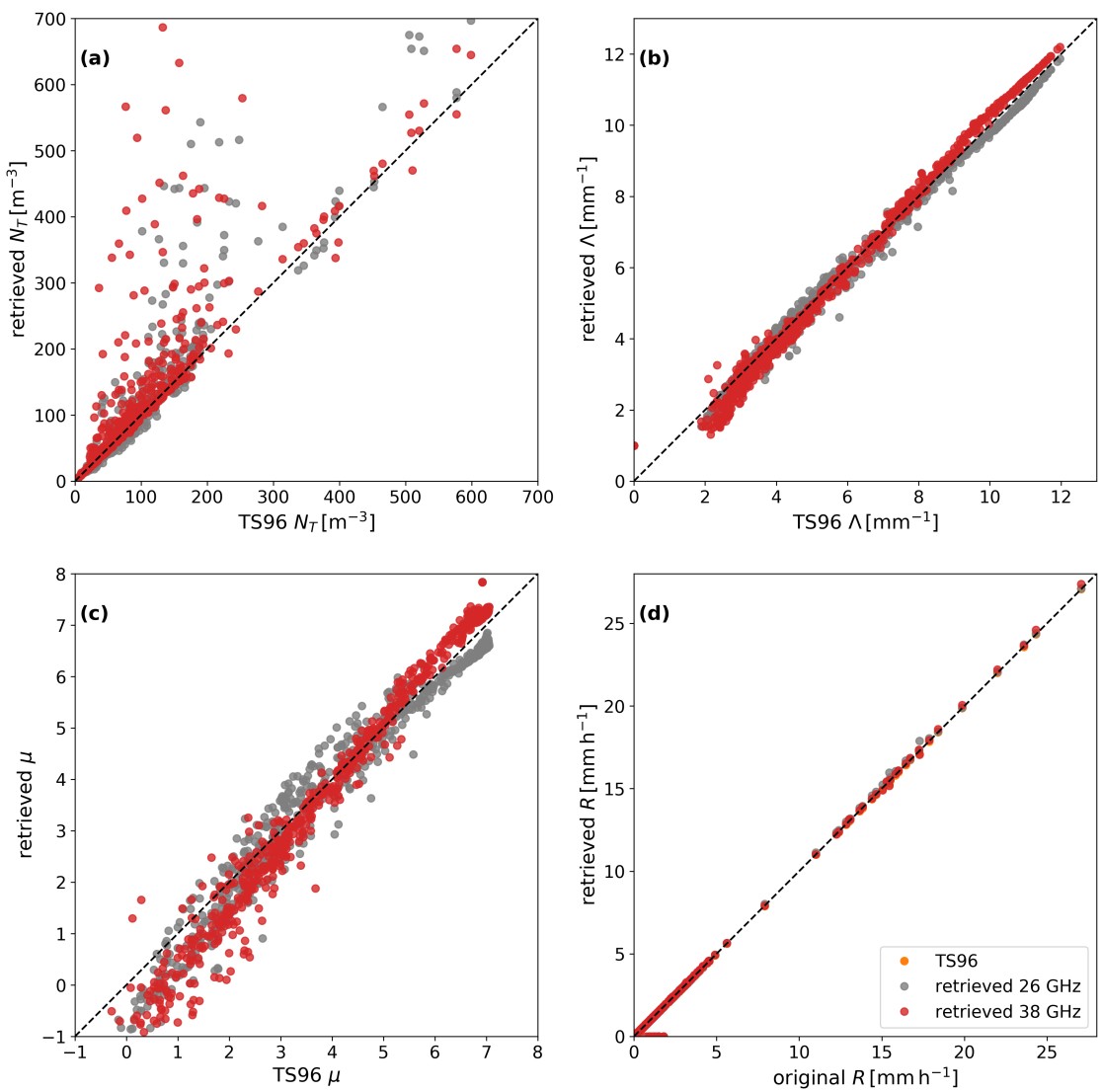

**Figure 8.** Gamma distribution parameters (a–c) and rainfall rates (d) retrieved from a simulation based on an event measured in the Ardèche region on 27 October 2013. The retrievals are performed using the three-parameter numerical microwave method and the TS96 method is shown as a reference. The dotted lines indicate a one-on-one relation.

**Table 2.** Statistics of rainfall intensity $R$ relevant to the accuracy and precision of the retrieval for one event and for the whole dataset using the two parameter method based on the Wageningen disdrometer dataset. All statistics are in comparison with the rainfall intensity directly calculated from the original DSD and normalized with respect to the median of the original measured rainfall intensity.

| | 15 GHz | 26 GHz | 32 GHz | 38 GHz |
|---|---|---|---|---|
| Event 1 (mean: 7.81 mm h$^{-1}$, median: 5.46 mm h$^{-1}$) | | | | |
| MOR | −0.0000 | 0.0084 | 0.0080 | 0.0083 |
| MAD | 0.0175 | 0.0251 | 0.0218 | 0.0239 |
| 95AD | 0.2510 | 0.1603 | 0.1473 | 0.2182 |
| fail | 0.0048 | 0.0000 | 0.0048 | 0.0048 |
| 9 months (mean: 1.43 mm h$^{-1}$, median: 0.69 mm h$^{-1}$) | | | | |
| MOR | 0.0003 | 0.0003 | 0.0001 | 0.0001 |
| MAD | 0.0105 | 0.0169 | 0.0154 | 0.0143 |
| 95AD | 0.3433 | 0.2563 | 0.2364 | 0.2509 |
| fail | 0.0184 | 0.0167 | 0.0171 | 0.0173 |

of the retrieval is an order of magnitude lower (higher MAD and 95AD), while the accuracy is actually higher (lower MOR) when compared with the Ardèche dataset.

## 5.1 Differences between two or three parameter retrievals

We summarize the differences in accuracy and precision between the two-parameter and the three-parameter method in Table 3. The analyses are all performed with respect to a 38-GHz dual-polarization retrieval (the third microwave link variable is the differential propagation phase). In addition, the number of failed retrievals (no solution at all) was 1.7 % when using only 2 moments, whereas there where no failed retrievals using three parameters within the filtered datset. This indicates that there is at least some advantage to using 3 moments.

However, the differences in accuracy and precision of the retrieval between the three-parameter and two-parameter retrieval as measured by a range of integer moments is small and in many cases the two-parameter retrieval proved to be more reliable. Especially the number of sub-millimeter rain drops is severely overestimated by using the three-parameter method, as shown in Fig. 10a. Fig. 10b also demonstrates that there is little difference in precision between the two-parameter and three-parameter method for any diameter class and even a slight advantage for the two-parameter method. Therefore, the addition of a third microwave link variable does not improve the retrieval (in many cases it actually harms the retrieval) and is unnecessary. Aside from needing one less measured moment, the two-parameter retrieval is also orders of magnitude faster. This is because the numerical part is univariate and therefore the dimensionality of the problem is reduced and also more efficient root finding

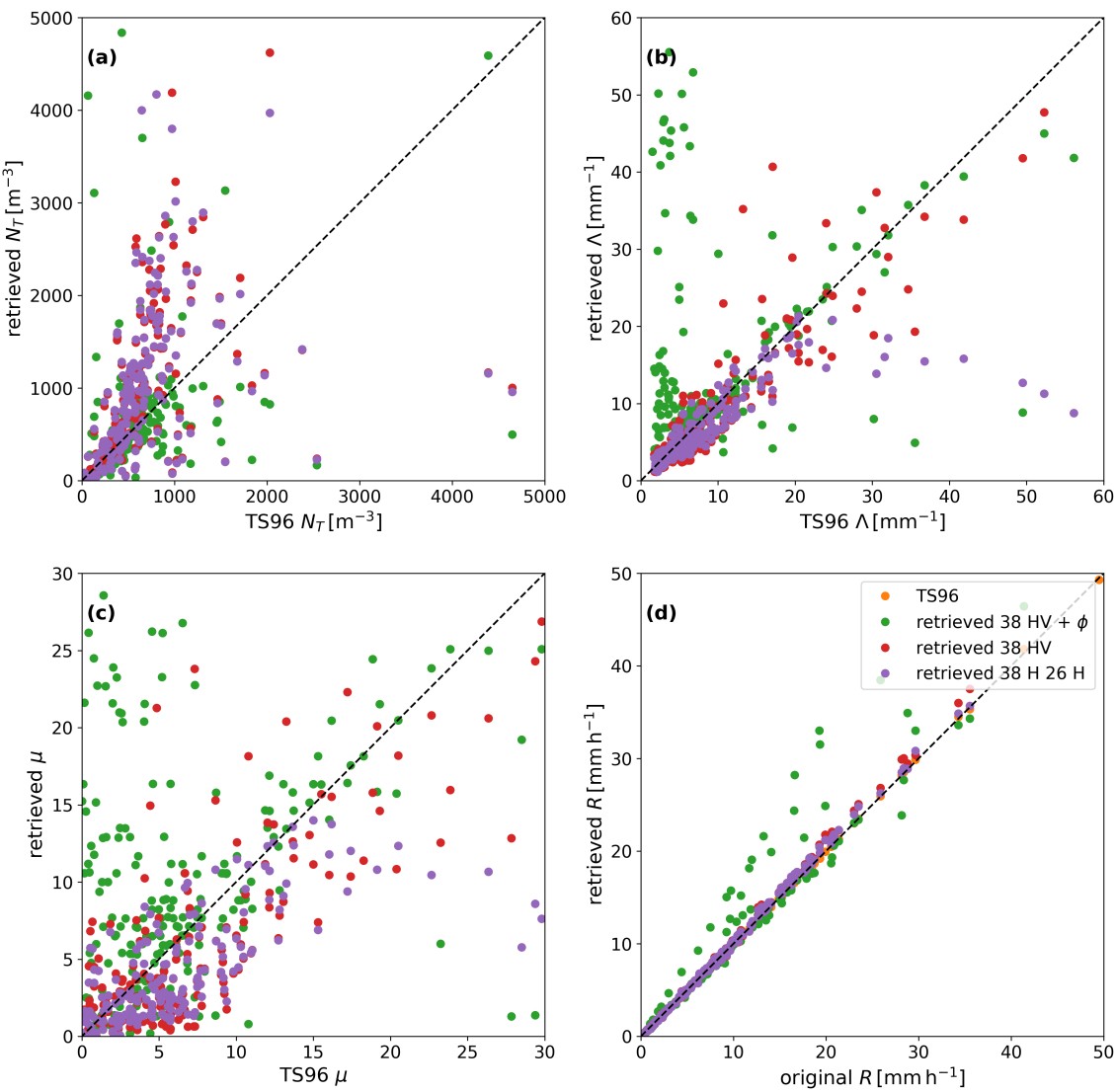

**Figure 9.** Gamma distribution parameters (a–c) and rainfall rates (d) retrieved from the event at Wageningen on 27 July 2015 based on attenuations derived from disdrometer measurements. Three different retrieval techniques are used: using the two attenuations and the phase difference at 38 GHz; using only the two attenuations at 38 GHz and a predetermined $\mu - \Lambda$ relation; using two attenuations at 38 GHz and 26 GHz and a predetermined $\mu - \Lambda$ relation. Data points which do not meet the filtering criteria described in Section 2.2 are excluded. The dotted lines indicate a one-on-one relation.

**Table 3.** Statistics of integer statistical moments $M_i$ relevant to the accuracy and precision of the retrieval based on disdrometer data for all 9 months with both types of retrievals at 38 GHz. All statistics are in comparison with the moments directly calculated from the original measured DSD and normalized with respect to the median of the original measured moments.

| | M0 | M1 | M2 | M3 | M4 | M5 | M6 |
|---|---|---|---|---|---|---|---|
| Mean | $196\,\mathrm{m^{-3}}$ | $155\,\mathrm{mm\,m^{-3}}$ | $144\,\mathrm{mm^2\,m^{-3}}$ | $163\,\mathrm{mm^3\,m^{-3}}$ | $236\,\mathrm{mm^4\,m^{-3}}$ | $467\,\mathrm{mm^5\,m^{-3}}$ | $1292\,\mathrm{mm^6\,m^{-3}}$ |
| Median | $127\,\mathrm{m^{-3}}$ | $101\,\mathrm{mm\,m^{-3}}$ | $89\,\mathrm{mm^2\,m^{-3}}$ | $88\,\mathrm{mm^3\,m^{-3}}$ | $95\,\mathrm{mm^4\,m^{-3}}$ | $113\,\mathrm{mm^5\,m^{-3}}$ | $149\,\mathrm{mm^6\,m^{-3}}$ |
| 2 parameters (failure ratio: 0.0173) | | | | | | | |
| MOR | 0.0639 | 0.0178 | -0.0009 | -0.0029 | 0.0001 | 0.0013 | 0.0018 |
| MAD | 0.2475 | 0.1379 | 0.0700 | 0.0324 | 0.0134 | 0.0086 | 0.0217 |
| 95AD | 1.9351 | 1.1617 | 0.6710 | 0.3692 | 0.3712 | 1.1472 | 3.8329 |
| 3 parameters (failure ratio: 0.0000) | | | | | | | |
| MOR | $-0.0118$ | $-0.0114$ | $-0.0085$ | $-0.0050$ | $-0.0048$ | $-0.0008$ | $-0.0009$ |
| MAD | 0.2849 | 0.1852 | 0.1091 | 0.0540 | 0.0174 | 0.0148 | 0.0381 |
| 95AD | 9.2275 | 6.1031 | 3.8252 | 2.0165 | 0.6672 | 2.5173 | 7.2813 |

methods other than gradient-based methods can be used (such as Brent's method). We do not need the workaround for local minima either, which is computationally very inefficient.

Because these results show that a three-parameter retrieval provides little added value above a two-parameter retrieval and because the two-parameter retrievals are far less computationally intensive than three-parameter retrievals we will restrict ourselves to two-parameter retrievals in the remainder of this paper.

### 5.2 Dependence on link frequency

In order to determine the effect of the carrier frequencies of the links on the accuracy and precision of the retrieval we perform two-parameter DSD retrievals at many different frequencies and calculate MOR, MAD and 95AD for the third order moment of the retrieval compared with the third order moment directly calculated from the measured DSD. We consider both dual-polarization retrievals with frequencies ranging from 10 GHz to 45 GHz (with steps of 1 GHz) as well as dual-frequency retrievals using every combination between 10 and 45 GHz. This range contains the bulk of microwave link frequencies found in typical communication networks. The results are shown for dual-polarization in Fig. 11 and for dual-frequency in Fig. 12. We can see in Fig. 11 that the accuracy and precision is high for all frequencies. However, the accuracy is pessimal for frequencies between 22 and 34 GHz. Similarly, MAD is largest around 25 GHz and 95AD is largest around 17 GHz. Therefore, for an optimal retrieval those intermediate frequencies should be avoided. Fig. 12 shows that the accuracy and precision of dual-frequency retrievals is highest when both frequencies are high. The difference between the two frequencies does not seem to matter much; when the two frequencies are far apart the precision and frequency are actually slightly lower. Predictably, there

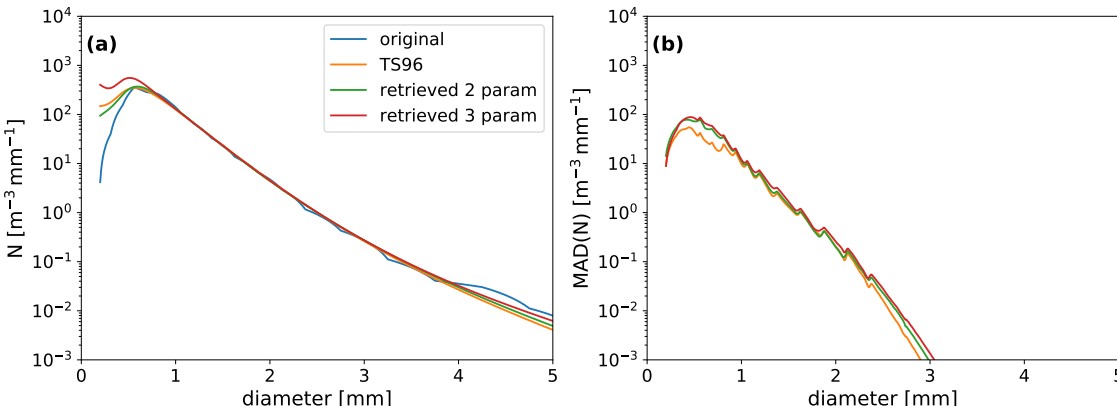

**Figure 10.** (a) Mean over 9 months of DSD retrievals from simulated microwave link data based on disdrometer data from the Wageningen experiment using three methods: the analytical method of TS96, a numerical approach based on horizontal and vertical attenuation at 38 GHz (two-parameter retrieval) and a numerical approach including both attenuations and the phase difference at 38 GHz (three-parameter retrieval). (b) Median absolute deviations per diameter interval for the aforementioned retrievals. The colors refer to the same retrieval method as in the first panel.

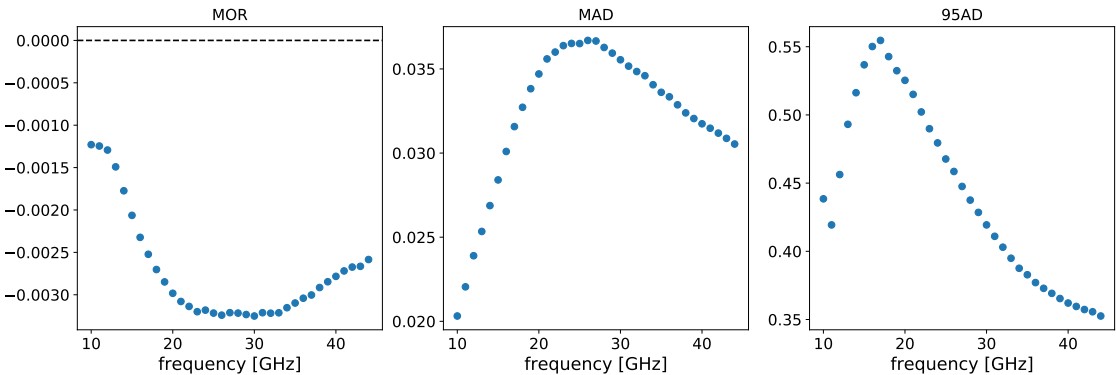

**Figure 11.** MOR, MAD and 95AD of the third order moment of the DSD estimated using a two-moment dual-polarization retrieval as a function of carrier frequency based on disdrometer data. All statistics are normalized with respect to the median of the moment of the original measured DSD.

are no solutions found when the frequencies are exactly the same. It should be noted that for this simulation, the effect of noise is not taken into account. It is expected that this would influence the retrieval the most when the frequencies are close together.

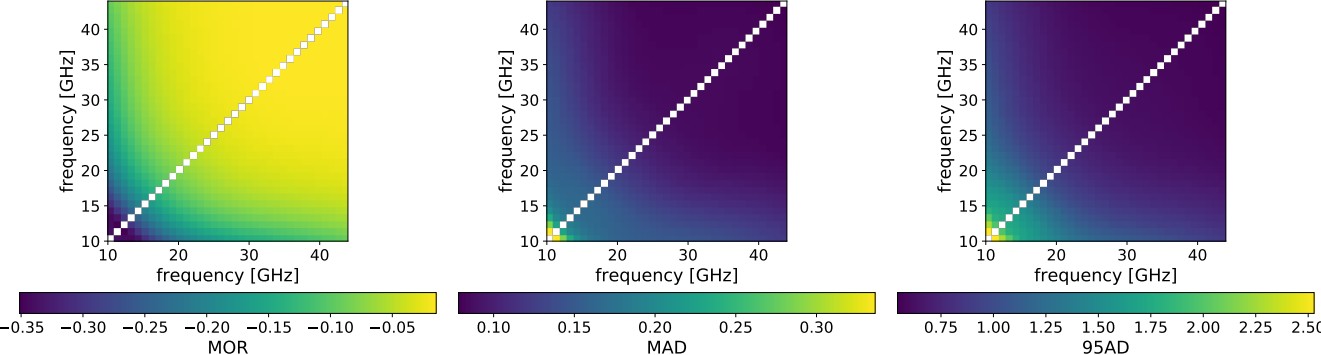

**Figure 12.** MOR, MAD and 95AD of the third order moment of the DSD estimated using a two-moment dual-frequency retrieval as a function of the two carrier frequencies based on disdrometer data. All statistics are normalized with respect to the median of the moment of the original measured DSD.

### 5.3 Sensitivity to attenuation bias

Because our retrieval algorithm uses ratios of attenuations as input, it is important that a reliable baseline power level is
established from which to calculate the attenuations. To assess the sensitivity of the retrieval technique to inaccuracies in the
baseline (dry) power level, we perform the two-moment retrievals based on the simulated attenuations from the disdrometer
measurements in Wageningen but with an (equal) offset added to all input attenuations. Figure 13 shows the resulting DSDs
averaged over 9 months for attenuation offsets between 0 and 5 dB and a path length of 2.2 km. For this analysis we chose two
combinations of links that we will also later use for the actual link measurements: a dual-polarization retrieval at 38 GHz and
a dual-frequency retrieval at 26 GHz and 38 GHz. We can see in Fig. 13a that the addition of an offset to the attenuation leads
to an overestimation below 2 mm and an underestimation above 2 mm for the dual-polarization retrieval. The introduction of
an offset has no effect at 2 mm and the effects are largest below 1 mm. The effects for a dual-frequency retrieval are quite
different, as can be seen in Fig. 13b. There is an overestimation at all diameters. The overestimation is smallest around 1 mm
and increasing towards higher and lower diameters. Overall, the mean bias is larger than for dual-polarization retrievals and
the shape of the DSD is especially sensitive to small offsets in the base power level.

### 5.4 Sensitivity to power quantization error

Aside from systematic measurement bias as discussed in Section 5.3, there can also be measurement limitations that affect
the precision of the attenuation measurements. Because the retrieval method relies on ratios of attenuations we expect that
the retrieval is highly sensitive to such limitations as well. In practice, when processing link attenuation data from operational
telecommunication networks, the power quantization error of the analog-to-digital conversion completely overshadows any

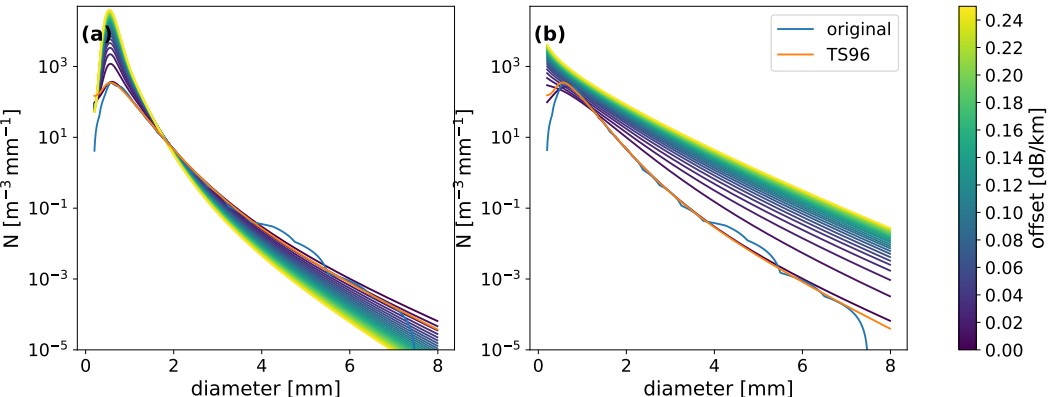

**Figure 13.** DSDs of the two-moment dual-polarization (38 GHz) (a) and dual-frequency (38 GHz and 26 GHz) (b) retrievals of the 9-month Wageningen disdrometer dataset with an offset in the attenuations of 0 – 5 dB.

instrumental error in the analog detector. Therefore, we will focus our analysis here exclusively on such quantization errors. Consequently, we do not have to make any assumptions about the instrumental precision of any particular transceiver model.

We have applied several different magnitudes of rounding to the attenuations calculated from the disdrometer data and performed the retrieval on the rounded attenuations for the complete 9-month dataset. The results are shown in Table 4. Common attenuation levels in operational networks are 0.1 dB, 0.5 dB and in some cases 1 dB. It is clear then, that the effect of quantization on the performance of the retrieval method is significant. Especially the number of nonconverging retrievals (from 1.7% to 66%) limits the prospective of successful application to current networks. The MAD calculated from the remaining successful retrievals is an order of magnitude higher than without quantization applied (see Table 2). The systematic bias is more than two orders of magnitude higher, but still limited (MOR = 4.3% at 0.1 dB). To achieve non-convergence ratios of less than 10% a quantization of 0.001 dB or less is required, which is makes this not achievable with current generation operational networks. It should also be noted that taking into account the quantization error in the analysis favors the dual-frequency method over the dual-polarization method. This can be attributed to the steeper slope of the attenuation-ratio–$\mu$ relationship within the band of common DSD shapes as shown in Fig. 5.

## 6 Experimental link retrieval

Using the double-moment retrieval method we estimated the DSDs from actual link measurements of the Wageningen setup. The baseline power level of the links showed considerable fluctuations over the course of the measurement period. Therefore, it was not feasible to perform retrievals for the entire 9-month dataset. We selected the event of 27 July 2015 (see Fig. 14), because the power levels of the links in the period surrounding this event showed relatively little fluctuations. We determined a suitable constant baseline power level calibrated for this event. The measured attenuation after subtracting a baseline for this

**Table 4.** Statistics of rainfall intensity $R$ relevant to the accuracy and precision of the retrieval for the whole dataset using the two parameter dual-polarization method (38 GHz) and different quantization levels based on the Wageningen disdrometer dataset. All statistics are normalized with respect to the median of the original measured rainfall intensity.

| quantization | 0.001 dB | 0.005 dB | 0.01 dB | 0.05 dB | 0.1 dB | 0.5 dB | 1.0 dB |
|---|---|---|---|---|---|---|---|
| dual-polarization (38 GHz) | | | | | | | |
| MOR | −0.0031 | −0.0091 | −0.0114 | 0.0026 | 0.0433 | 0.7408 | 1.2148 |
| MAD | 0.0207 | 0.0375 | 0.0482 | 0.1036 | 0.1708 | 0.8108 | 1.7778 |
| 95AD | 0.2651 | 0.2967 | 0.3289 | 0.5877 | 0.8629 | 3.6008 | 11.1076 |
| fail | 0.0810 | 0.2032 | 0.2892 | 0.5569 | 0.6602 | 0.5827 | 0.4133 |
| dual-frequency (26/38 GHz) | | | | | | | |
| MOR | −0.0122 | −0.0171 | −0.0240 | −0.0490 | −0.0549 | 0.0240 | 0.1524 |
| MAD | 0.0350 | 0.0449 | 0.0528 | 0.0805 | 0.1060 | 0.3284 | 0.5973 |
| 95AD | 0.2789 | 0.2955 | 0.3148 | 0.4324 | 0.5367 | 1.0648 | 1.7650 |
| fail | 0.0659 | 0.1539 | 0.2058 | 0.3465 | 0.3918 | 0.3807 | 0.3097 |

event is shown in Fig. 14 and compared with the path-average attenuation derived from the disdrometer measurements; the retrieval results are given in Fig. 15. It should be noted that for this experiment we used the analog detector voltage directly fed into a data logger with a 13-bit ADC and no further quantization applied, so—unlike for operational networks—we expect quantization effects to be small compared to instrumental errors.

The resulting DSD is very similar in shape to that obtained in the simulations, with overestimations especially at smaller diameters, but with the general shape of the DSD preserved (not shown). Closer inspection reveals that the bias and scatter compared to the original DSD are actually up to two orders of magnitude higher than in the simulations, as can be seen in Table 5 when compared to Table 3. The scatter as indicated by the MAD is about one order of magnitude higher than the simulations across all moments. However, when taking the 95AD as measure of scatter, the order of magnitude is the same. There are more intervals with no solution at all when compared to the simulations. These correspond with ratios of observables that are outside the range of the forward model (see Fig. 5). The ratios for the dual-polarization retrieval are illustrated in Fig. 14b. These time intervals with extremely low or high ratios between attenuations mostly (but not always) occur when the rain intensity is low and thus other sources of signal variability are more dominant. Overall, the dual-polarization and dual-frequency retrievals have a similar performance in this case. However, using two frequencies instead of two polarizations leads to a higher accuracy for low order moments (up to 4th), but lower accuracy for higher order moments. Similar to the simulated retrievals based on the disdrometer dataset there is an overestimation of the $N_T$-parameter and underestimation of the $\mu$-parameter. There is also an overestimation of the $\Lambda$-parameter. Especially the dual-polarization retrieval yields some outliers which overestimate $\mu$ but underestimate $\Lambda$. No plausible solutions for the retrieval were obtained when using the phase

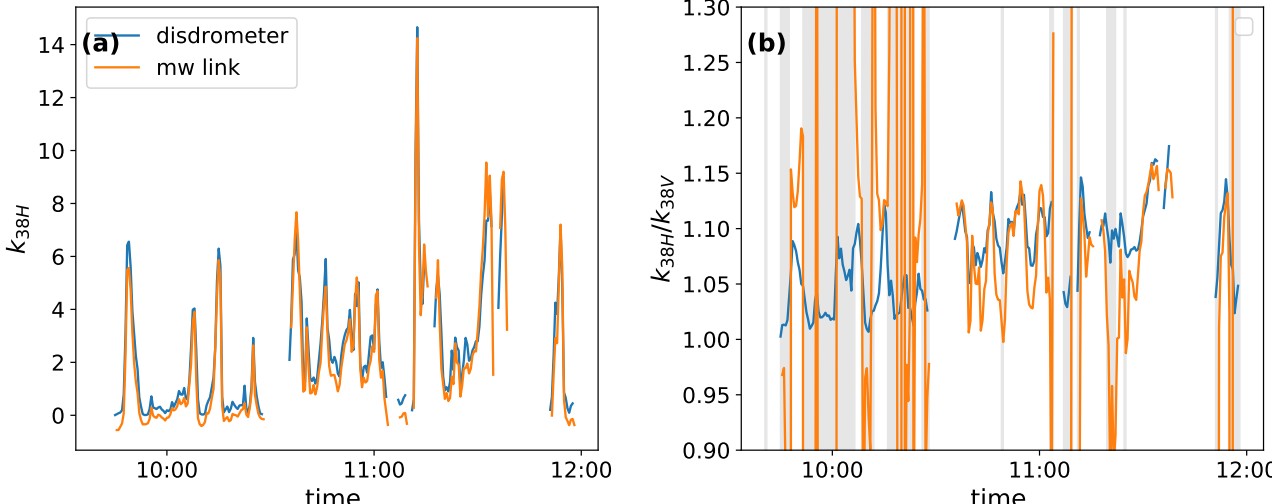

**Figure 14.** a) Link attenuation at 38GHz with vertical polarization. Time series are shown for link measurements (orange) as well as attenuation computed from disdrometer data (blue) for the event of 27 July 2015. b) Ratios of the attenuations of the horizontally and vertically polarized signal, from link and disdrometer data. Shaded areas indicate intervals where the ratio is outside of the solvable range for the dual-polarization retrieval (1.00 – 1.25; see Fig. (5)).

difference instead of one of the attenuations or with a triple variable retrieval. Very few intervals showed convergence at all with this configuration.

## 7   Discussion

### 7.1   Feasibility in practice

Constraints on the feasibility of the proposed methods in practice fall into three broad categories: availability of multiple link signals on the same path; quality of the available signals; and real-time processing speed.

The use of a three-moment retrieval means that three moments on the same path need to be available. This is rare in commercial networks; therefore this method is most readily applicable to dedicated research networks. There are several different combinations of moments to choose from. However, in our approach we focused on the combination of a horizontal attenuation, vertical attenuation and phase difference at the same carrier frequency. This allows the use of a single set of antennae for all three moments, allowing for the use of a more compact and less expensive device (such as the device that was used in our test setup).

The second concern is with regard to the quality and reliability of the signal. In order to apply the method in practice it is essential that a baseline (no rain) signal is accurately determined. Because the method relies on the ratios of attenuations,

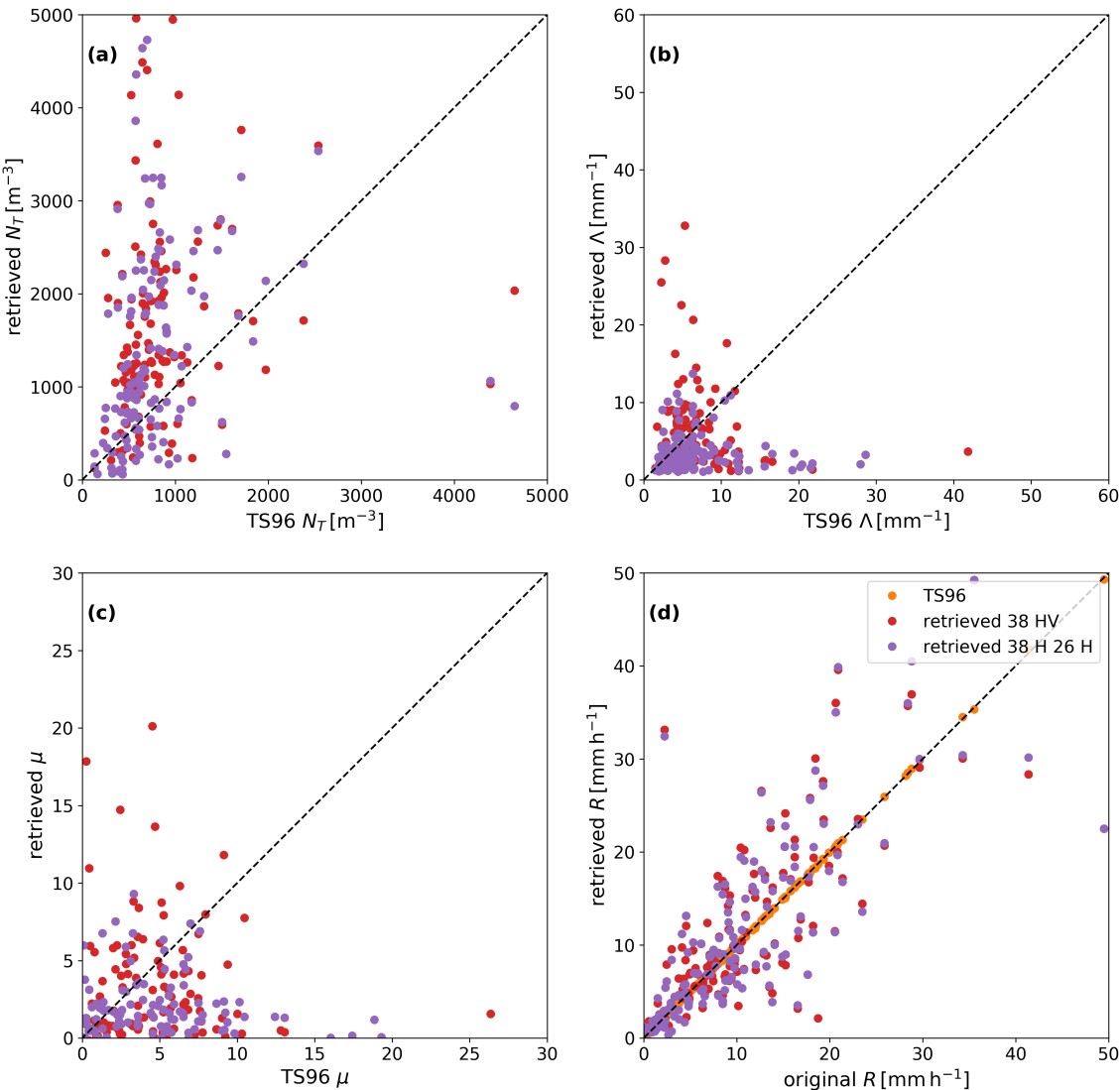

**Figure 15.** Gamma distribution parameters (a–c) and rainfall rates (d) retrieved from the event of 27 July 2015 as measured by the 38 GHz instrument using horizontal and vertical polarization and the 26 GHz instrument using horizontal polarization. The TS96 results are derived from the five disdrometers for the same event. Data points which do not meet the filtering criteria described in Section 2.2 are excluded. The dotted lines indicate a one-on-one relation.

**Table 5.** Statistics of accuracy and precision of the retrieval of integer moments of the DSD for the event of 27 July 2015 using actual link data from two different combinations of links. All statistics are normalized with respect to the median of the moments of the disdrometer data.

| | M0 | M1 | M2 | M3 | M4 | M5 | M6 |
|---|---|---|---|---|---|---|---|
| Mean | $475\ \mathrm{m}^{-3}$ | $471\ \mathrm{mm\,m}^{-3}$ | $550\ \mathrm{mm}^2\,\mathrm{m}^{-3}$ | $765\ \mathrm{mm}^3\,\mathrm{m}^{-3}$ | $1289\ \mathrm{mm}^4\,\mathrm{m}^{-3}$ | $2688\ \mathrm{mm}^5\,\mathrm{m}^{-3}$ | $6994\ \mathrm{mm}^6\,\mathrm{m}^{-3}$ |
| Median | $476\ \mathrm{m}^{-3}$ | $453\ \mathrm{mm\,m}^{-3}$ | $500\ \mathrm{mm}^2\,\mathrm{m}^{-3}$ | $607\ \mathrm{mm}^3\,\mathrm{m}^{-3}$ | $804\ \mathrm{mm}^4\,\mathrm{m}^{-3}$ | $1306\ \mathrm{mm}^5\,\mathrm{m}^{-3}$ | $2523\ \mathrm{mm}^6\,\mathrm{m}^{-3}$ |
| **38 GHz H, V (failure ratio: 0.3190)** | | | | | | | |
| MOR | 1.4768 | 0.7821 | 0.3955 | 0.1078 | $-0.0135$ | $-0.1448$ | $-0.1738$ |
| MAD | 0.5684 | 0.4039 | 0.2825 | 0.1871 | 0.2120 | 0.5800 | 1.1188 |
| 95AD | 2.4817 | 1.8452 | 1.2116 | 0.8446 | 1.3568 | 3.0648 | 7.0133 |
| **26 GHz H, 38 GHz H (failure ratio: 0.2857)** | | | | | | | |
| MOR | 0.7707 | 0.2506 | 0.0236 | -0.0521 | 0.0026 | 0.1951 | 0.3954 |
| MAD | 1.1068 | 0.6947 | 0.4442 | 0.2269 | 0.2304 | 0.5302 | 1.1808 |
| 95AD | 2.7860 | 2.0012 | 1.3678 | 1.0405 | 0.9715 | 2.8380 | 6.6522 |

small deviations in the baseline determination can result in large deviations in the retrieval (and even non-convergence). No such problem exists in principle with regard to the phase difference; it is independent of any power baseline. However, phase difference on its own is not sufficient for the retrieval. To have a chance at a successful retrieval the baseline needs to be as invariant as possible. Where it is not, the variability should be accurately modeled and predicted from auxiliary measurements. In our own preliminary attempts we found our instruments lacking in stability. In particular the clinging of drops to the antenna cover (as described in van Leth et al., 2018) seems an intractable problem. However, as was also described in that paper, we found that a former commercial microwave link had a much stabler baseline and furthermore the effect of wet antennas was much more manageable for that particular device. This provides a hopeful perspective for the application of this method to commercial networks, in particular if data could be logged with high precision (Chwala et al., 2012). However, when using data from such commercial networks, the quantization of the signal plays a much bigger role. We have seen that the typical quantization levels applied to the data collected by the network operators are insufficient to allow reliable DSD retrievals. The only way to apply such retrievals to currently operational unmodified link networks consistently is to install dedicated data-loggers at selected link locations to read out the analog signal directly. This may not be feasible. Therefore, a more high-resolution default storage of link received power level by future models of transceiver units would be needed before wide-scale application can take place.

The third constraint is only relevant when real-time processing is required. The three-moment method is relatively wasteful with computing cycles because of the repeated reinitialization of the root finding process. We might expect this to become a bottleneck. However, in its current implementation the processing of 9 months of data from one link requires roughly three

395     hours of wall time on a high-end desktop workstation. This is while utilizing 10 CPU cores simultaneously. Such a setup is therefore expected to be able to process a network of more than 2000 links in real-time. Nevertheless, reducing this computational load would make real-time retrievals more feasible on low-end (embedded) hardware as well. We already found that the specific programmatic implementation of the root-finding algorithm can make an order of magnitude difference in computation time. The use of pre-computed lookup tables may help to bring down computation time in a real-time setting. The two-moment

method requires far less computation (e.g. processing the same 9 months of data with the same workstation requires only 5 minutes), which makes it far more suitable for real-time processing.

## 7.2   Caveats

There are a number of caveats to our methods which could influence the interpretation of the results: firstly, we use a threshold of 50 drops per disdrometer to filter out low quality measurements before calculating the mask or $\mu$-$\Lambda$ fit. How high this

number should be is debatable (e.g. Uijlenhoet et al. (2006)). A threshold that is too low might allow for many erroneous measurements that are not representative of rain. On the other hand, a threshold that is too high might result in too many reasonable measurements being rejected. This might mean the results are less statistically representative and possibly biased towards high rain intensities. The number used here is therefore a compromise and the resulting relationships do change somewhat depending on the threshold chosen. A similar consideration applies to the choice to filter on a per-instrument basis

instead of on the basis of the total drop count. Filtering on a per-instrument basis makes it more likely that all instruments where measuring correctly. However, it does bias the measurements towards more homogeneous rainfall. Considering that the employed disdrometers were located relatively close together and therefore that their measurements are strongly correlated, we accepted this potential bias.

    Another consideration is the use of the mask itself. The mask is determined on the basis of the measured disdrometer data.

We then use this mask in, among others, the retrieval of the DSD from the disdrometer-derived simulated variables. This could potentially lead to a retrieval procedure that is biased towards these particular circumstances and therefore yields more accurate retrievals in this simulation than would be representative for a general application. Nevertheless, this data is never used as input for the root-finding procedure itself. It is only used a posteriori to assess whether the results fall into a plausible range of values. Another potential issue is the use of the predetermined $\mu$-$\Lambda$ relationship in the case of the two-parameter retrieval. In this case,

the relationship determined on the basis of the disdrometer measurements is used directly as an assumption in the retrieval of these parameters from the disdrometer measurements. However, this is justified by the fact that both the $\mu$-$\Lambda$ relationship and the mask are determined from the total of all 9 months of disdrometer measurements, not from the specific event in question. It should also be noted that for the retrievals from the Ardèche dataset we have also used both the mask and the $\mu$-$\Lambda$ relationship determined from the Wageningen dataset.

The third consideration is the underlying assumption that the gamma distribution is an adequate representation of the actual DSD and that the untruncated gamma distribution is applicable throughout the diameter domain. It is on the basis of this assumption that we treat the values of $\mu$ and $\Lambda$ derived from the TS96 method of moments as the correct parameter values and the DSD resulting from that as representative of the actual DSD. The gamma distribution has a non-zero value up to

positive infinity. Meanwhile, there are both physical and instrumental cutoffs to the maximum drop size that can occur. This would suggest that a truncation should be included in the expression of the gamma distribution. However, the truncation of the gamma distribution at high diameters is not relevant in this case because we can see in Figs. 6a, 10a and 13 that the gamma distribution corresponding to the DSDs under consideration tends to zero (or $< 10^{-2}\,\mathrm{m}^{-3}\,\mathrm{mm}^{-1}$) before the instrumental cutoff. We can also observe from those figures that the systematic deviation between the measured (interpolated) DSD and the DSD obtained from the method of moments is small, which suggests that the gamma distribution is a valid approximation for the 30 second aggregation interval that has been used here. Instrumental cutoff at the small end of the diameter scale is relevant in this case, but the effects of this on higher order moments is minute. Regardless, since the attenuation-/phase-based retrieval in this case is limited to three parameters, it is not possible to include a cutoff there. A retrieval using even more signals might make this possible, but this would further limit the practical applicability of this method.

## 8    Conclusions and outlook

Using simulated link data we have shown that a DSD retrieval on the basis of multiple microwave link variables can be successful and accurate, but only when precise high-resolution records of rain-induced attenuation are available. This was confirmed when applied on actual link data, where baseline variations prohibited accurate DSD retrievals. Both the use of dual-polarization and dual-frequency retrievals are feasible. However, the use of dual-polarization is less sensitive to systematic inaccuracies in the base power level while being more sensitive to quantization errors than the use of dual-frequency links. Simulated retrievals using a variety of frequencies show that, at least between 10 GHz and 45 GHz, the accuracy and precision of the retrievals is very high for all frequency combinations. Therefore, the frequency chosen for a dual-polarization retrieval is not very important. Nevertheless, when a choice is available, our simulations indicate that frequencies at the lowest end or the highest end of the range are preferred. Furthermore, at the lowest end of the frequency range there is less attenuation in general and therefore a smaller difference in attenuation. This could more easily be obscured by noise or quantization effects. Therefore, the higher end of the tested frequencies are optimal. For dual-frequency retrievals, bias and random error are an order of magnitude higher than for the dual-polarization retrievals when no input error is assumed. If a dual-frequency retrieval is attempted, both frequencies should be as high as possible to minimize the bias and random error.

In our field experiment we tested a dual-frequency retrieval using 26 GHz and 38 GHz as well as a dual-polarization retrieval at 38 GHz. Both retrievals produced some reasonable results for a selected summer event where other attenuating atmospheric phenomena where not present, but there where many intervals within this event where no solution was obtained at all. The feasibility of the retrieval depends very much on a stable base power level, which was not guaranteed in our experiment. The Nokia link (former commercial link) is promising in this respect, because it was much stabler and less sensitive to e.g. temperature fluctuations and antenna wetting. A follow-up experiment using two commercial links is planned for the near future.

Using phase differences in addition to attenuations is feasible in the simulations. However, in practice these measurements are not accurate enough to yield meaningful solutions. In most instances no convergence was obtained. We have also shown

that using three microwave link variables yields no improvements over a retrieval using only two variables, which is also computationally faster and more readily applicable in operational settings. At least in comparable climatologies to those treated here, a predetermined $\mu$-$\Lambda$ relation suffices to determine the gamma DSD parameters from two attenuations.

A follow-up experiment using different microwave links of similar frequencies (preferably commercially available ones) is needed to determine if the base power level of commercial links is sufficiently stable for reliable continuous observations. A tally should also be done on the number of dual-polarized links in cellular communications networks to determine if it is feasible to retrieve spatial DSD information from such networks or whether this technique is only applicable to some individual link paths, either from commercial or research networks.

Another concern is the quantization of data from commercial link networks. As the difference between the attenuation of the horizontally polarized signal and the vertically polarized signal is often a fraction of a $\mathrm{dB}$ and data available from such networks are often rounded to 0.1, 0.5 or 1 $\mathrm{dB}$, this limits the applicability of this method severely. Therefore, the possibility of collecting higher resolution data from such instruments should be looked into.

*Data availability.* The underlying radar and disdrometer data employed to simulate the DSD dataset described in Section 2.1 is available through the HyMeX database (http://www.hymex.org). The link and disdrometer data from the Wageningen link experiment (described in Section 2.2) are publicly available through the 4TU data repository (https://doi.org/10.4121/uuid:1dd45123-c732-4390-9fe4-6e09b578d4ff).

*Author contributions.* This paper was conceived by TCvL, with some suggestions from RU. TCvL devised the algorithm, wrote the necessary
software to perform the retrievals and analyzed the results, with feedback by HL and RU. TCvL wrote the first draft and, after critical revision by all co-authors, wrote the final version.

*Competing interests.* The authors declare no competing interests.

*Acknowledgements.* The Ardèche dataset was kindly provided by Timothy H. Raupach. Funding was provided by the Netherlands Organi-
zation for Scientific Research (NWO-TTW; project number 11944).

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
