# Peer review of "Estimating raindrop size distributions using microwave link measurements: potential and limitations"

_Atmospheric Measurement Techniques, 2019_

## Referee Comment (RC1) · Anonymous Referee #1 · 14 May 2019

**Estimating raindrop size distributions using microwave link measurements**
by T. van Leth, H. Leijnse, A. Overeem and R. Uijlenhoet

**Summary:**
Two methods for retrieving drop size distributions (DSDs) from microwave link measurements are proposed. The methods are evaluated theoretically using simulated DSD fields as well as on real data using 5 disdrometers along a 2.2 km dual-frequency dual-polarization link located in Wageningen. The simulations show that in theory, both retrieval methods are feasible, although one of them is numerically more stable than the other. The application to real data appears more problematic. Retrieved DSDs were not necessarily reliable due to large measurement uncertainty and biases in the baseline attenuation and wet antenna attenuation.

**Assessment:**
Granted, some decent retrievals were obtained on carefully selected datapoints. However, Figure 15 speaks for itself. It shows that overall, there is a very poor agreement between the actual measured attenuation ratios (from the disdrometers) and the ones inferred from the links. The authors provide little explanation for this nor do they give numbers for the overall accuracy over the whole dataset. However, from the text, it is quite obvious that the overall reliability on real data remains very low. Because of that, I can not recommend publication at this point. If the goal is to show the practical limits of the method, then my suggestion to the authors would be to dig deeper and take advantage of their experimental framework to further test and validate the quality and feasibility of their retrievals over the whole dataset, including proper uncertainty analysis and recommendations for when and how to retrieve the DSDs.

**Recommendation:** major review

**1. Theoretical weaknesses in the retrieval methods:**
The first retrieval method (using 3 measurements) seems to be very unstable with lots of convergence issues. Even when the algorithm converges, the solution is not necessarily unique. This prompted the authors to add additional assumptions and constraints, such as a range of plausible values or the use of a bivariate probability distribution for mu and lambda derived from disdrometer data. The main problem with this approach is that you go from a physical, purely data driven retrieval to something that seems to strongly depend on model assumptions. The whole thing feels a bit arbitrary to me and it is unclear how much of the information in the measurements you still need/use when doing the retrievals.
The second model is less messy numerically speaking but heavily relies on the adequacy of the mu-lambda relationship. It's all fine in theory but there are several practical problems related to how microwave links operate that limit the usability of this method. The most important of them is noise/uncertainty in measured attenuation values (see next comment). In fact, the authors already acknowledge this in the paper by saying that it was not feasible to perform retrievals for the entire 9-month dataset. This is not a good sign. What's the point of having a method that you can't apply most of the time?

**2. Novelty:** Page 2: "To the best of our knowledge, no further research has been published regarding DSD estimations using microwave links.". Actually, there appears to be a conference proceeding by Berne and Schleiss (2009) at the 34th Conference on Radar Meteorology in Williamsburg that mentions the possibility to retrieve DSDs from dual-polarization links using the exact same technique (i.e., based

on the ratio of attenuations at H and V). Interestingly, their work never made it through peer-review and does not appear to have been published. My guess is that they faced the same practical problems.

**3. Lack of proper uncertainty analysis:**
You absolutely need to provide some form of confidence interval or lower/upper bounds on the retrieved mu and Nt values! This would help put things into perspective and provide the reader with more realistic expectations of what can be retrieved and under what circumstances. This can easily be done using the simulated DSDs and some basic assumptions about noise levels in microwave links.

4. **The accuracy and reliability of the DSD retrievals in operational links is not clear:**
The simulation experiments show that the retrieval methods work fine in theory. However, there are several practical problems that need to be investigated more carefully: The most important is related to the quantization of the power measurements. In commercial links, attenuation is usually measured in steps of 0.1 or 0.3 dB. Values are rounded up or down depending on the quantizer and this is done independently for each channel or polarization. Consequently, the measured attenuation ratio might be affected by a very large uncertainty. This effect can be simulated to get an idea of how it affects the retrievals. A simple calculation shows that a +/- 0.1 dB quantization noise on each channel is enough to ruin most retrievals for low to moderate rainfall rates. The second problem is wet antenna attenuation or more generally, any other form of bias in the baseline that affects the power level. This is partially explored in Sections 5.3 where the authors quantify the effect of measurement biases on the average DSD (over 9 months). The discussion in 7.1 also mentions some limitations for operational links. However, the text remains overly optimistic and evaluations based on climatological DSDs are insufficient to conclude anything about the instantaneous values. Please provide more details on this.

**5. The assessment is heavily focuses on weighted moments rather than the DSD itself:**
The current paper is very vague when it comes to assessing the accuracy on the retrieved DSDs. It puts a lot of emphasis on integrated moments such as rainfall rate. Also, an evaluation based on average DSDs over the entire even is not enough and I would like to see more details about performance on the actual, instantaneous retrieved DSDs (e.g., the mu and lambda values).

**6. Some graphs could be improved:**
The time series format used to illustrate the retrieved DSDs using different lines and colors is clearly not optimal. Often, colors overlap and the individual lines are hard to distinguish from each other. A scatterplot containing all estimated mu values versus the disdrometer reference together with some basic statistics would give a better overview. Alternatively, histograms or boxplots of mu, lambda and R could be used.

**Minor comments:**
- Figure 8 (and others): The scale for lambda seems wrong. The values should be in the same order of magnitude than mu (or even slightly larger). Please check!
- Page 13: *"We can also see that in several timesteps the μ and Λ parameters in the retrieval are several times higher than they are in the TS96 method, but that this does not result in a significantly different rain intensity"* This should not come as a surprise, as rainfall rate is heavily conditioned by the specific attenuation at these frequencies. The concentration parameter will compensate for a wrong DSD shape.
- Section 7.3: I would add the fact that the Gamma model itself may not be adequate at representing the actual DSD, especially at high temporal resolutions. This is probably more important than the truncation. Many previous studies have shown that, although they come relatively close, strictly speaking, many DSDs measured by disdrometers are not really gamma.

---

## Referee Comment (RC2) · Anonymous Referee #2 · 14 May 2019

**REVIEW REPORT**

Review of amt-2019-51-manuscript-version1

By Thomas C. van Leth, Hidde Leijnse, Aart Overeem, and Remko Uijlenhoet

Manuscript Title – Estimating raindrop size distributions using microwave link measurements

**GENERAL COMMENTS**

In the manuscript the Authors exploit the microwave links for the estimation of drop size distribution (DSD). The study include analysis based on simulated data and analysis conducted on real data collected by three collocated microwave links and four OTT Parsivel disdrometers. I think that the research topic is of high interest and have potentiality to improve the DSD knowledge and estimation, however in some part the paper is a bit hard to follow and confused. Furthermore some Figures should be done in a different way because now it is really hard to identify the differences among the different datasets. The Authors made a lot of different analysis and, in order to help the reader, more clarity and explanations are needed. I suggest a major revision and recommend the publication of the paper on the Atmospheric Measurement Techniques after addressing the following comments and suggestions.

**SPECIFIC COMMENTS**

1. Introduction, first paragraph: Regarding the use of "signal of opportunity" to retrieve precipitation, in the last decade some studies has been carried out to investigate also the usefulness of geostationary broadcast television satellite links. A reference also to this technique should be inserted in the Introduction section (such as Giannetti et al. 2017 and references therein.

   Giannetti, F., Reggiannini, R., Moretti, M., Adirosi, E., Baldini, L., Facheris, L., Andrea Antonini, Melani S, Bacci G., Petrolino A., Vaccaro, A. (2017). Real-time rain rate evaluation via satellite downlink signal attenuation measurement. Sensors, 17(8), 1864, doi: https://doi.org/10.3390/s17081864

2. Section 2.1, first paragraph: To help the reader to understand the advection-based temporal interpolation technique, can the Authors add few information regarding this technique? I understand that the DSD retrieval is based on the polarimetric radar data, but which is the role of disdromter data? How many disdrometers there are in the 20 km x 20 km area? Which is the location of the disdrometer? Which is the distance of the 2D interpolated DSD field from the radar?

3. Section 2.1, second paragraph: How do the Authors select the position of the transect? Does the latter choice has an impact on the results? The transect consist in 1x200 pixels, correct?

4. Section 2.1, third paragraph: Can the Authors quantify the impact of binning effect on the results? Basically, it would be useful to know which is the differences in terms of

attenuations and differential phase shifts considering DSD binned as Parsivel and DSD re-binned in regular diameter grid with dD = 0.1 mm. Knowing the latter information will help the reader to understand the impact of the binning on the results.

5. Section 3.1: I suggest to change the title of this subsection with "Theoretical background" or something similar. It not describe a new procedure but a well-known methodology to retrieve attenuation and specific phase shift from DSD.

6. Section 3.2, second-last line: "In order to prevent this we restrict the root finding algorithm to a limited range of parameter values". Which are these ranges? How did the Authors define them?

7. Page 8, first 2 lines: If I understand well the Authors basically change the first guess values until the method converges and finds a solution. Is it enough? I mean in this way the methods find a solution for all the DSDs? Which is the percentage of samples that do not have a solution?

8. Page 9: "We prefer this method because it is not based on gradients and therefore guaranteed to find a solution if it exists". Similar to comment #7, How many times the solution does not exist? Please provide a percentage.

9. Section 3.4, first two lines: "We test the capability of the methods to accurately retrieve DSDs and their associated statistical moments with two different datasets of **measured** drop size distributions". The Authors use also smilated DSD dataset. Correct? Please clarify

10. Section 3.4, first paragraph: Please put the TS96 abbreviation before, when the Tokay and Short (1996) method is cited for the first time. Furthermore if the Authors want to use this abbreviation to refer to the method of moment proposed by Tokay and Short (1996), please use it within all the text and Figure. In many Figures and in some part of the text the Authors referred to Tokay and Short (1996) method with "method of moments" and some times with "TS96". Chose one!

11. Section 3.4, first paragraph: It is not clear to me how the TS96 results are applied to "distinguish between cases where the gamma distribution is simply not a good fit for the measured DSD and cases where the retrieval itself is the cause for inaccuracies". Please clarify it.

12. Section 4.1, first 2 lines: Which input data are used for this "typical three-parameter retrieval"? Data from Ardèche dataset? Please explain

13. Section 4.1, second line: In Figure 7 there are different lines that refer to different frequencies, not only to the 38 GHz, why in the text the Authors refers only to the 38 GHz? Please explain

14. Section 4.1, third line: "$N_m$ is the originally measured DSD", is the world "measured" correct? If yes please clarify why the section title refer to simulated DSD and why in the previous line the Authors refer to simulations ("between the retrieved DSD and the original simulation procedure"). It is not clear to me.

15. Section 4.1: I don't understand this sentence "The difference in the total drop concentration is $\Delta N_T < 0.2 \cdot N_T$ in the first case, while the difference in the total rain intensity is $\Delta R < 0.03 \cdot R_o$.". Please clarify

16. Ficure 7c: Why the Authors do not puts the differences between original DSD and TS96 DSD?

17. Section 4.2: Can the Author identify the type of the two events (26 November 2012 and 27 October 2013)? Stratiform or Convective?

18. Figure 8:

    a. Most of the time it is not possible to see the TS96 line (blue line). Please provide another method to visualize the results such as a scatterlot between TS96 and the 26GHz or 38Ghz retrievals.

    b. Please put the legend in a position that not cover the data

    c. the method of moments is TS96? If yes please for clarity refer always to the same acronym/name within the text. The latter is valid for all the Figures. Try to use for all the figures the same color for the same dataset. Example: blue line is for "original" in Figure7 and for "method of moments" in Figure 8

    d. I am not confident with your advection-based temporal interpolation technique used to retrieve DSD from radar data, however usually the DSD retrieval techniques from radar data provide mu and lambda. Why the Authors do not use this data (the so called "original data" in Figure 7 and 9) to compare the obtained results at 26 GHz and 38Ghz?

19. Section 5, first sentence: Here the retrieval from disdrometer data are compared with disdrometer data. Correct? Please specify. Please explain clearly in each figure which is the reference ("true") line/dataset

20. Figure 9:

    a. See comment 18c

    b. See comment 18a

    c. Why here do the Authors insert the retrieved 15 Ghz and 32 Ghz and in Figure 8 there aren't? Please explain

21. Table 2: The MOR MAD and 95AD have been computed between retrieved DSD at different frequencies and the TS96 values? Please clarify. If yes, why the Authors do not use the R obtained directly from disdrometer DSD?

22. Section 5.1, first paragraph: Which are the percentage of failed retrieval for the two-parameter and the three-parameter methods? Here the Author provide the differences

between the two percentages (1.7%), however I think that is useful also to have the two percentage values.

23. Figure 10:

    a. See comment 18c

    b. See comment 18a

    c. Add the label on x-axis

    d. In figure10a the reference value is the TS96, while in Figure 10b and c the reference is the original DSD, correct? Please add this information in the text

24. Table 3: The MOR MAD and 95AD values are obtained comparing the retrieval with the original or with the TS96? Please clarify

25. Figure 11: Can the Authors explain why the 3-parameter retrieval overestimates the small drops with respect to 2-param retrieval?

26. Section 5.2, second line: Please clarify the two dataset used to compute the MOR, MAD and 95AD. Disdrometer based R and 2-parameter retrieved R?

27. Figure 14: Please provide a better explanation of the figure. what is a)? and b)?

28. Section 6:  I believe that this is the most important part of the paper, therefore all the analysis and results have to be explained with more detail and clarity.

29. Section 6, line n.10: A lot of different analysis have been done in the paper, therefore to help the reader please identify which is the Table to be compared with Table 4.

30. Section 6: "Nevertheless, at the important higher order moments related to e.g. liquid water content, rain rate, kinetic energy and radar reflectivity the bias is around 7 % for the dual-polarization retrieval". The bias between….? It is not clear to me the 2 dataset used to compute the bias. Please clarify

31. Figure 16:

    a. In Figure 16b) also the R from original DSD ca be added

    b. please provide the label for x-axis

**TECHNICAL CORRECTIONS**

1. Section 2.2, first line: erase the word "second"
2. Figure 2: please put the legend outside the plot area, otherwise it covers some lines
3. Figure 5: Probably the Author can eliminate this figure and add the lamda-mu relation in Figure 4b. It is just a suggestion

4. Page 23, line n. 2: "Because" should be uppercase
5. Page 23, line n. 4: "It" should be lowercase
6. Section 7.2, first line: "Firstly" should be lowercase

---

## Referee Comment (RC3) · Anonymous Referee #3 · 16 May 2019

**General comments:**

The manuscript concerns estimation of drop size distribution (DSD) from attenuation of radiowaves at different frequencies/polarizations. The focal point of the manuscript lays in validation of proposed methods by numerical simulations, nevertheless applicability of the method is demonstrated also during a single rainfall event on attenuation measurements obtained from an experimental setup with dual polarized 38 GHz and horizontally polarized 26 GHz microwave links and array of disdrometers. The topic is relevant and the methodology is scientifically valid. The manuscript is also well structured and very well written.

My major concern is in applicability of the presented approach on real attenuation measurements obtained from commercial microwave links (CMLs), which is where the proposed methods have the highest potential. The DSD estimation is thoroughly tested on simulated attenuation observations, which are to my understanding ideal (not perturbed by any errors). This should be clearly stated probably already in the Method section because it is very important for interpretation of the results. The authors are apparently aware of different limits and pitfalls when it comes to application of the method on CML data, nevertheless the discussion of these limitations and pitfalls could (should) be more specific. This is important, because some of the conclusions based on numerical experiments (e.g. that dual frequency method is insensitive to difference between frequencies) are to my understanding only valid for ideal CMLs with high precision and accuracy. Detailed sensitivity or uncertainty analysis which would enable to quantify effect of inaccuracies in attenuation measurement on the efficiency of the proposed methods is probably out of the scope of this study, nevertheless the manuscript would clearly benefit from more robust discussion of the results in the context of real attenuation measurements from CML networks. This issue is further discussed in the specific comments.

Despite my concern regarding applicability of the method in real CML setting I consider the authors' work as a valuable contribution to DSD research and research related to exploitation of CMLs for environmental monitoring and believe that authors can address this issue by relatively minor revisions.

**Specific comments:**

P1L8 – 9 Abstract: Isn't the accuracy of the method highly dependent also on precision of the measurements as noted in the Conclusions or more general on accuracy of identified rainfall induced attenuation?

P3L10 – 14: Why do you use transect of 20 km when typical length of CMLs operated at frequencies 15 – 38 GHz (and esp. 26 – 38 GHz) is substantially shorter? Moreover, you later demonstrate the method on 2.2 km CML. Could you indicate the reason for

simulating CML over entire length of the field?

P7L2: The reasoning should probably refer to eq. 14 instead of Eq. 11 – 13 to apply not only for dual polarization setting but also for dual frequency setting. Furthermore, variables used in eq. 15 are defined in eq. 14 and not in Eqs. 11 – 13.

P13 – 15: Please comment on spells with no results which can be seen on Fig. 10 (and later also on fig. 16). Are they due to not identified parameters, or due to measurement outages? Please, comment on these 'outages' also on P20.

P18L12 – P19L5 (the whole section): It is not clear if the offset is applied to both frequencies resp. polarizations. If yes, then the whole analysis is not much informative. And what if bias evolves in time differently for both polarizations? This might be the case e.g. due to wet antenna attenuation when droplets on the surface of antennas start to transform themselves into rivulets (see e.g. Mancini et al., 2019).

As noted in the Conclusion section, stability of baseline level is crucial in practice for utilizing the proposed technique. In practice, the feasibility of your DSD estimation approach will be probably very much sensitive to ratio between rainfall induced attenuation and other sources of attenuation which cannot be easily identified, i.e. it is reasonable to expect that method will be applicable only to CMLs relatively sensitive to rainfall (longer CMLs, higher frequencies) where effect of wet antenna attenuation and limited quantization is not so much pronounced. Given this, it might be much more informative to investigate sensitivity of the methods to precision of CMLs, e.g. simple averaging might very well simulate quantization of CMLs.

I fully acknowledge that detail sensitivity or uncertainty analysis on the whole dataset might be sufficient for stand-alone work and it is out of the scope of this investigation, however, performing such analysis on a subset of data (e.g. the event used for demonstration on real data) would be probably sufficient to enable discussing applicability of the method in real CML network in more specific manner (see the next comment to the P23L1 – 11).

In any case, the limitations of recent sensitivity analysis should be addressed either in the section itself or within discussion section. It would be also valuable to i) refer to typical values of baseline offset and ii) to include into discussion of this issue also wet antenna effect. Finally consider presenting results in dB/km. This would make them applicable also to other link lengths.

P23L1 – 11: The issues discussed in this paragraph are of crucial importance for application of the method in a real CML network. It would make sense to discuss in here also quantization of CMLs (which is raised at the end of the Conclusions without previous deeper discussion). Also discussion on wet antenna attenuation might be deeper, e.g. to which extend we can expect it will differ for two frequencies or polarizations? Finally, the results presented in the section Dependence on link frequency should be discussed in a view of inaccuracy of real CML measurements. It is likely that by lower frequencies (e.g. 17 GHz) only relatively long CMLs will have sufficient sensitivity to rainfall (and thus sufficient precision) to be suitable for DSD estimation. Similarly, insensitivity of DSD estimation on an offset between frequencies (now presented as one of the conclusions) does not apply for real data with limited precision. The attenuation difference will be for typical CML frequencies and lengths at least for light and moderate rainfalls below quantization of the CML records.

P23L33 – P24L7: The second paragraph of the Caveats section lacks exactness of the previous text with vague formulations like 'potentially more serious', 'disdrometer measurements is used more directly', or 'somewhat in favor of'. Detailed investigation and quantitative assessment of discussed issues is probably out of the scope of this manuscript, nevertheless, authors might consider reformulating the second paragraph to provide more specific reasoning why and to which extend can the mask influence the results and overall applicability of the proposed methods.

P23L29: typo, 'too' instead of 'to'

P25L15 – 17: The limited precision of CMLs due to quantization is a very important

limitation which should be discussed in more detail already in the Discussion section (see comment to P23L1 − 11).

**Some further questions I would be curious about (no need to answer):**

- The three parameter method which uses as an input three attenuation measurements ($k_{1,2,3}$) provided less accurate DSD estimate than two-parameter method. Have you tried to apply two-parameter method on different combinations of attenuations ($k_1$ / $k_2$, $k_1/k_3$, $k_2/k_3$) and use the redundant parameter estimates for improving the estimation accuracy (e.g. identifying outliers)?

- There is probably some autocorrelation in parameters of DSD function, have you thought about using autocorrelation structure of DSD observations to constrain the optimization procedure?

––––––––––––––––––––––––

---

## Referee Comment (RC4) · Anonymous Referee #3 · 16 May 2019

This is a full reference to the manuscript I refer to in my comment (RC3).

**References:**

Mancini, A., Lebrón, R.M., Salazar, J.L., 2019. The Impact of a Wet S-Band Radome on Dual-Polarized Phased-Array Radar System Performance. IEEE Trans. Antennas Propag. 67, 207–220. https://doi.org/10.1109/TAP.2018.2876733

---

## Author Response (AR1)

**Review 1**

**Summary:**

*Two methods for retrieving drop size distributions (DSDs) from microwave link measurements are proposed. The methods are evaluated theoretically using simulated DSD fields as well as on real data using 5 disdrometers along a 2.2 km dual-frequency dual-polarization link located in Wageningen. The simulations show that in theory, both retrieval methods are feasible, although one of them is numerically more stable than the other. The application to real data appears more problematic. Retrieved DSDs were not necessarily reliable due to large measurement uncertainty and biases in the baseline attenuation and wet antenna attenuation.*

**Assessment:**

*Granted, some decent retrievals were obtained on carefully selected datapoints. However, Figure 15 speaks for itself. It shows that overall, there is a very poor agreement between the actual measured attenuation ratios (from the disdrometers) and the ones inferred from the links. The authors provide little explanation for this nor do they give numbers for the overall accuracy over the whole dataset. However, from the text, it is quite obvious that the overall reliability on real data remains very low. Because of that, I can not recommend publication at this point. If the goal is to show the practical limits of the method, then my suggestion to the authors would be to dig deeper and take advantage of their experimental framework to further test and validate the quality and feasibility of their retrievals over the whole dataset, including proper uncertainty analysis and recommendations for when and how to retrieve the DSDs.*

**Recommendation:** *major review*

The reviewer focuses exclusively on the quality of the experimental retrieval and the lack of in-depth analysis of these practical retrievals as grounds for major revisions. However, the practical retrievals are not the main topic of this paper and serve only illustrative purposes. The flaws in the practical setup have little to do with the retrieval method itself or the simulation results which form the primary message of the paper. The flaws with the experimental setup are known and described by the authors. It is also known and described that these flaws are not a given for operational CMLs and may be exclusive to the instruments used here. Therefore further investigation of practical retrievals using this method is warranted, but not within this experimental framework. The suggestion to dig deeper into the experimental data collected is therefore not helpful and further analysis over the whole experimental dataset would not yield meaningful results.

The authors are aware of the need for more practical validation and do intend to set up a new experimental framework in the near future that would hopefully avoid the flaws of the previous one (which has long since been dismantled). That would be out of scope for this paper, though. We will gladly take the critique pertaining to the simulated results to heart and improve on that in a revision.

If the criticism is that if the practical validation here is of so little value, then we should not include it at all, then that would be a valid point and something we might consider. However, we think that the inclusion of these preliminary experimental results does serve an illustrative purpose.

**1. Theoretical weaknesses in the retrieval methods:** *The first retrieval method (using 3 measurements) seems to be very unstable with lots of convergence issues. Even when the algorithm converges, the solution is not necessarily unique. This prompted the authors to add additional assumptions and constraints, such as a range of plausible values or the use of a bivariate probability distribution for mu and lambda derived from disdrometer data. The main problem with this approach is that you go from a physical, purely data driven retrieval to something that seems to strongly depend on model assumptions. The whole thing feels a bit arbitrary to me and it is unclear how much of the information in the measurements you still need/use when doing the retrievals.*
*The second model is less messy numerically speaking but heavily relies on the adequacy of the mu-lambda relationship. It's all fine in theory but there are several practical problems related to how microwave links operate that limit the usability of this method. The most important of them is noise/uncertainty in measured attenuation values (see next comment). In fact, the authors already acknowledge this in the paper by saying that it was not feasible to perform retrievals for the entire 9- month dataset. This is not a good sign. What's the point of having a method that you can't apply most of the time?*

There appears to be little actionable comment here. These aspects of the methods are already described and discussed. The original intent was to attempt a purely physical retrieval based on three parameters. This turned out to be unstable even with simulated data and could only be made stable using these constraints. We fully recognize how unsatisfying this may be, but we find that all the more reason to report this. The two-parameter retrieval does not have these problems and, as already mentioned, the lack of proper retrievals with our experimental setup does not provide a conclusive case against the feasibility of the method.

**2. Novelty:** *Page 2: "To the best of our knowledge, no further research has been published regarding DSD estimations using microwave links.". Actually, there appears to be a conference proceeding by Berne and Schleiss (2009) at the 34th Conference on Radar Meteorology in Williamsburg that mentions the possibility to retrieve DSDs from dual-polarization links using the exact same technique (i.e., based on the ratio of attenuations at H and V). Interestingly, their work never made it through peer-review and does not appear to have been published. My guess is that they faced the same practical problems.*

The authors where not aware of this when the article was submitted. We do believe that the fact that this methodology has been attempted before (apparently unfruitfully) and not published (Rincon & Lang also promised a follow up in their paper that never materialized) is all the more reason to thoroughly investigate the method and publish the results, even if the method proves ultimately infeasible.

**3. Lack of proper uncertainty analysis:** *You absolutely need to provide some form of confidence interval or lower/upper bounds on the retrieved mu and Nt values! This would help put things into*

*perspective and provide the reader with more realistic expectations of what can be retrieved and under what circumstances. This can easily be done using the simulated DSDs and some basic assumptions about noise levels in microwave links.*

We prefer to avoid making assumptions about the noise levels in microwave links that we cannot back up and we believe an analysis based on such assumptions would only be misleading. However, we do know the typical quantization errors found in operational networks, which are very likely far more limiting that the instrumental noise, and have an analysis of their effect to the retrieval in the revision.

**4. The accuracy and reliability of the DSD retrievals in operational links is not clear:**
*The simulation experiments show that the retrieval methods work fine in theory. However, there are several practical problems that need to be investigated more carefully: The most important is related to the quantization of the power measurements. In commercial links, attenuation is usually measured in steps of 0.1 or 0.3 dB. Values are rounded up or down depending on the quantizer and this is done independently for each channel or polarization. Consequently, the measured attenuation ratio might be affected by a very large uncertainty. This effect can be simulated to get an idea of how it affects the retrievals. A simple calculation shows that a +/- 0.1 dB quantization noise on each channel is enough to ruin most retrievals for low to moderate rainfall rates. The second problem is wet antenna attenuation or more generally, any other form of bias in the baseline that affects the power level. This is partially explored in Sections 5.3 where the authors quantify the effect of measurement biases on the average DSD (over 9 months). The discussion in 7.1 also mentions some limitations for operational links. However, the text remains overly optimistic and evaluations based on climatological DSDs are insufficient to conclude anything about the instantaneous values. Please provide more details on this.*

Using the simulations to assess the effect of quantization error on the performance of the retrieval would indeed be a valuable addition to the analyses performed here and would give a better idea of the performance in operational networks. We have expanded on this in the revision.

**5. The assessment is heavily focuses on weighted moments rather than the DSD itself:**
*The current paper is very vague when it comes to assessing the accuracy on the retrieved DSDs. It puts a lot of emphasis on integrated moments such as rainfall rate. Also, an evaluation based on average DSDs over the entire even is not enough and I would like to see more details about performance on the actual, instantaneous retrieved DSDs (e.g., the mu and lambda values).*

The statistical moments where chosen because they can be directly compared with the original DSDs without assuming a gamma distribution in the original DSD. Therefore the effect that the assumption of a gamma distribution in the retrieval has is included in the metrics. This would not be the case when applying the metrics to mu and lambda values. The evaluation based on average DSDs over the entire event or 9 month period provides information about the bias present in the instantaneous DSDs. We have added a new graph in figure 10b that shows the MAD of the DSD calculated over the 9 month period. This provides a good metric for the random error present in the instantaneous DSDs.

**6. Some graphs could be improved:** *The time series format used to illustrate the retrieved DSDs using different lines and colors is clearly not optimal. Often, colors overlap and the individual lines are hard to distinguish from each other. A scatterplot containing all estimated mu values versus the disdrometer reference together with some basic statistics would give a better overview. Alternatively, histograms or boxplots of mu, lambda and R could be used.*

We have decided to replace most of the time series graphs with scatterplots in the revision and made some visual adjustments to the remaining time series to make them hopefully more clear.

**Minor comments:**
*- Figure 8 (and others): The scale for lambda seems wrong. The values should be in the same order of magnitude than mu (or even slightly larger). Please check!*

The y axis in these figures is given as $\Lambda^{-1}$. This is incorrectly labeled in Figure 8 (but correct in others). This choice was made in order to make better use of the visual space in the graph. However, because this is a potential source of confusion, we decided not to use $\Lambda^{-1}$ anymore in the new graphs.

*- Page 13: "We can also see that in several timesteps the $\mu$ and $\Lambda$ parameters in the retrieval are several times higher than they are in the TS96 method, but that this does not result in a significantly different rain intensity" This should not come as a surprise, as rainfall rate is heavily conditioned by the specific attenuation at these frequencies. The concentration parameter will compensate for a wrong DSD shape.*

It is still worth mentioning. We have slightly reformulated and corrected some of the statements here.

*- Section 7.3: I would add the fact that the Gamma model itself may not be adequate at representing the actual DSD, especially at high temporal resolutions. This is probably more important than the truncation. Many previous studies have shown that, although they come relatively close, strictly speaking, many DSDs measured by disdrometers are not really gamma.*

This is a good point and we have revised this part of the discussion to include both the effect of truncation and the choice of the gamma distribution itself.

**Review 2**

*In the manuscript the Authors exploit the microwave links for the estimation of drop size distribution (DSD). The study include analysis based on simulated data and analysis conducted on real data collected by three collocated microwave links and four OTT Parsivel disdrometers. I think that the research topic is of high interest and have potentiality to improve the DSD knowledge and estimation, however in some part the paper is a bit hard to follow and confused. Furthermore some Figures should be done in a different way because now it is really hard to identify the differences among the different datasets. The Authors made a lot of different analysis and, in order to help the reader, more clarity and explanations are needed. I suggest a major revision and recommend the publication of the paper on the Atmospheric Measurement Techniques after addressing the following comments and suggestions.*

We appreciate the interest and comments of the reviewer. The reviewer points out that some parts of the text and captions are unclear or ambiguous. The reviewer also points out that the presentation of the results in the figures is in some cases not effective at conveying the information in a clear manner. We acknowledge these concerns and therefore would like to address these concerns in a revised version of the manuscript where will clear up some confusing text, add extra explanations and represent some of the results in a different visual form.

The reviewer also provides a list with specific comments, which we will address in a pointwise manner below.

**SPECIFIC COMMENTS**

*1. Introduction, first paragraph: Regarding the use of "signal of opportunity" to retrieve precipitation, in the last decade some studies has been carried out to investigate also the usefulness of geostationary broadcast television satellite links. A reference also to this technique should be inserted in the Introduction section (such as Giannetti et al. 2017 and references therein.*

*Giannetti, F., Reggiannini, R., Moretti, M., Adirosi, E., Baldini, L., Facheris, L., Andrea Antonini, Melani S, Bacci G., Petrolino A., Vaccaro, A. (2017). Real-time rain rate evaluation via satellite downlink signal attenuation measurement. Sensors, 17(8), 1864, doi: https://doi.org/10.3390/s17081864*

This is a closely related development and it would be indeed be good to mention in the introductory paragraph for completeness sake. We have added a sentence pointing out this development in the revision.

*2. Section 2.1, first paragraph: To help the reader to understand the advection-based temporal interpolation technique, can the Authors add few information regarding this technique? I understand that the DSD retrieval is based on the polarimetric radar data, but which is the role of dis-*

*dromter data? How many disdromteters there are in the 20 km x 20 km area? Which is the location of the disdrometer? Which is the distance of the 2D interpolated DSD field from the radar?*

The specifics concerning the technique and the underlying dataset can be found in the papers referenced in this paragraph (Raupach & Berne 2016, 2017; De Vos et al, 2018). We have made some clarifications in the revised text regarding this issue.

*3. Section 2.1, second paragraph: How do the Authors select the position of the transect? Does the latter choice has an impact on the results? The transect consist in 1x200 pixels, correct?*

The choice of the transect in this case was rather arbitrary; It is the center line.

*4. Section 2.1, third paragraph: Can the Authors quantify the impact of binning effect on the results? Basically, it would be useful to know which is the differences in terms of attenuations and differential phase shifts considering DSD binned as Parsivel and DSD rebinned in regular diameter grid with dD = 0.1 mm. Knowing the latter information will help the reader to understand the impact of the binning on the results.*

When the scattering cross sections are only known at the Parsivel bin center diameters, the retrievals are not possible because too much information is lost. A detailed analysis of the exact effect of different binning strategies is beyond the scope of this paper.

*5. Section 3.1: I suggest to change the title of this subsection with "Theoretical background" or something similar. It not describe a new procedure but a well-known methodology to retrieve attenuation and specific phase shift from DSD.*

The word "new" is not included in the title. This suggestion would not be an improvement.

*6. Section 3.2, second-last line: "In order to prevent this we restrict the root finding algorithm to a limited range of parameter values". Which are these ranges? How did the Authors define them?*

The range of values that was used in the eventual analysis is equal to the mask that is used to constrain the values. This is mentioned in P8L11.

*7. Page 8, first 2 lines: If I understand well the Authors basically change the first guess values until the method converges and finds a solution. Is it enough? I mean in this way the methods find a solution for all the DSDs? Which is the percentage of samples that do not have a solution?*

We have included detailed information about the convergence failure rates for different retrieval variants in the revised paper. The number of failed retrievals using this method is 0 (At least for the time intervals with a minimum number of drops as explained in the text).

*8. Page 9: "We prefer this method because it is not based on gradients and therefore guaranteed to find a solution if it exists". Similar to comment #7, How many times the solution does not exist?*

*Please provide a percentage.*

We have included the fraction unsolved retrievals as an additional metric to assess the performance in the revision. For the whole 9 months it is 1.7% for the time intervals with the minimum number of drops.

*9. Section 3.4, first two lines: "We test the capability of the methods to accurately retrieve DSDs and their associated statistical moments with two different datasets of measured drop size distributions". The Authors use also smilated DSD dataset. Correct? Please clarify*

This is correct. There is one simulated and one measured dataset. We have changed the line to clarify this.

*10. Section 3.4, first paragraph: Please put the TS96 abbreviation before, when the Tokay and Short (1996) method is cited for the first time. Furthermore if the Authors want to use this abbreviation to refer to the method of moment proposed by Tokay and Short (1996), please use it within all the text and Figure. In many Figures and in some part of the text the Authors referred to Tokay and Short (1996) method with "method of moments" and some times with "TS96". Chose one!*

We have revised the text and used TS96 for all references to this method.

*11. Section 3.4, first paragraph : It is not clear to me how the TS96 results are applied to "distinguish between cases where the gamma distribution is simply not a good fit for the measured DSD and cases where the retrieval itself is the cause for inaccuracies". Please clarify it.*

If the gamma distribution is not a good fit for the measured DSD, then the results from the TS96 method would deviate significantly from the measured DSD. If this is not the case, but the retrieved DSD does deviate significantly from the measured DSD, then the retrieval itself is the main cause of inaccuracy in the DSD.

*12. Section 4.1, first 2 lines: Which input data are used for this "typical three-parameter retrieval"? Data from Ardèche dataset? Please explain*

Section 4 heading: "Validation using simulated DSD". We have clarified this in the running text as well in the revision.

*13. Section 4.1, second line: In Figure 7 there are different lines that refer to different frequencies, not only to the 38 GHz, why in the text the Authors refers only to the 38 GHz? Please explain*

This is an unintended inconsistency. We have revised to use the same set of frequencies in both figures and text.

*14. Section 4.1, third line: "N m is the originally measured DSD", is the world "measured" correct? If yes please clarify why the section title refer to simulated DSD and why in the previous line the Authors refer to simulations ("between the retrieved DSD and the original simulation procedure").*

*It is not clear to me.*

"Originally measured DSD" refers here to the simulated DSD in the case of the Ardeche dataset. We have reworded this as "Original DSD".

*15. Section 4.1: I don't understand this sentence "The difference in the total drop concentration is $\Delta N_T < 0.2 \cdot N_T$ in the first case, while the difference in the total rain intensity is $\Delta R < 0.03 \cdot R_o$ .". Please clarify*

This means the relative difference in terms of drop concentration is 20 % while the relative difference in terms of rain intensity is 3 %.

*16. Ficure 7c: Why the Authors do not puts the differences between original DSD and TS96 DSD?*

This is an unintended error in the graph. We have corrected this.

*17. Section 4.2: Can the Author identify the type of the two events (26 November 2012 and 27 October 2013)? Stratiform or Convective?*

Both events are based on data collected in the South-East of France in the season with the annual rainfall maximum. They can both be classified as orographic or convective events. The second event is more spatially heterogeneous than the first with a decorrelation distance of 2.8 km vs 11 km at 30 s accumulation intervals. See also De Vos et al. (2018) (full reference in the manuscript). We have expanded our description of the dataset with this information in the revised paper.

*18. Figure 8:*

*a. Most of the time it is not possible to see the TS96 line (blue line). Please provide another method to visualize the results such as a scatterlot between TS96 and the 26GHz or 38Ghz retrievals.*

The lack of readability of the time-series plots has been noted by other reviewers as well. We have replaced most of the time-series plots with scatterplots in the revision.

*b. Please put the legend in a position that not cover the data*

We will try to pay more attention to this with the revised figures.

*c. the method of moments is TS96? If yes please for clarity refer always to the same acronym/name within the text. The latter is valid for all the Figures. Try to use for all the figures the same color for the same dataset. Example: blue line is for "original" in Figure7 and for "method of moments" in Figure 8*

Method of moments refers here to TS96. We have reworded to be more consistent. We have tried to be consistent in the use of line colors, however figure 8 seems to have escaped our attention. In the revised graphs we have made use of the same colors consistently.

*d. I am not confident with your advection-based temporal interpolation technique used to retrieve DSD from radar data, however usually the DSD retrieval techniques from radar data provide mu and lambda. Why the Authors do not use this data (the so called "original data" in Figure 7 and 9) to compare the obtained results at 26 GHz and 38Ghz?*

The DSDs retrieved from this method come in the form of diameter bins not as $\mu$ and $\Lambda$ values. Therefore, the binned DSD is the original and mu and lambda need to be derived from that. The procedure is described in Raupach & Berne (2016) and De Vos et al. (2018). (Full references are included in the manuscript)

*19. Section 5, first sentence: Here the retrieval from disdrometer data are compared with disdrometer data. Correct? Please specify. Please explain clearly in each figure which is the reference ("true") line/dataset*

Correct. In all figures the reference dataset is referred to in the legend as "original".

*20. Figure 9:*

*a. See comment 18c*

See our answer there.

*b. See comment 18a*

See our answer there.

*c. Why here do the Authors insert the retrieved 15 Ghz and 32 Ghz and in Figure 8 there aren't? Please explain*

This is an unintended inconsistency. We have revised our paper to use the same set of frequencies in both figures.

*21. Table 2: The MOR MAD and 95AD have been computed between retrieved DSD at different frequencies and the TS96 values? Please clarify. If yes, why the Authors do not use the R obtained directly from disdrometer DSD?*

We do use the values of R obtained directly from the disdrometer DSD rather than the TS96 values.

*22. Section 5.1, first paragraph: Which are the percentage of failed retrieval for the two-parameter and the three-parameter methods? Here the Author provide the differences between the two per-*

centages (1.7%), however I think that is useful also to have the two percentage values.

This is the only instance where the failure rate is mentioned. We agree that this is not enough and we have added this as an additional metric to MOR, MAD and 95AD in all relevant tables in the revision.

*23. Figure 10:*

*a. See comment 18c*

See our answer there.

*b. See comment 18a*

See our answer there.

*c. Add the label on x-axis*

We have replaced these figures entirely in the revision so this is no longer relevant.

*d. In figure10a the reference value is the TS96, while in Figure 10b and c the reference is the original DSD, correct? Please add this information in the text*

This is correct. In the revision these graphs have been replaced by scatterplots and this should now be a lot clearer.

*24. Table 3: The MOR MAD and 95AD values are obtained comparing the retrieval with the original or with the TS96? Please clarify*

They are compared with the original. We have made this more clear in the caption.

*25. Figure 11: Can the Authors explain why the 3-parameter retrieval overestimates the small drops with respect to 2-param retrieval?*

We currently do not a have a solid explanation for this and prefer not to speculate about it.

*26. Section 5.2, second line: Please clarify the two dataset used to compute the MOR, MAD and 95AD. Disdrometer based R and 2-parameter retrieved R?*

Correct. This is true for all results in section 5. We have clarified this in the text.

*27. Figure 14: Please provide a better explanation of the figure. what is a)? and b)?*

Fig 14a shows dual-polarization retrievals, while fig 14b shows dual-frequency retrievals. We have clarified this in the caption.

*28. Section 6: I believe that this is the most important part of the paper, therefore all the analysis and results have to be explained with more detail and clarity.*

Actually, we do not consider this the most important part of the paper. Our experimental link setup proved insufficient for an experimental validation of the method because of the high instability of the system which is worse than actual operational CMLs. We plan to do a follow up investigation using a new experimental setup consisting of only formerly operational CMLs. This will provide a more thorough experimental validation, but is out of scope for now. The experimental results here provide only a proof of concept.

*29. Section 6, line n.10: A lot of different analysis have been done in the paper, therefore to help the reader please identify which is the Table to be compared with Table 4.*

Compared to Table 3. We have clarified this in the revision.

*30. Section 6: "Nevertheless, at the important higher order moments related to e.g. liquid water content, rain rate, kinetic energy and radar reflectivity the bias is around 7 % for the dual-polarization retrieval". The bias between....? It is not clear to me the 2 dataset used to compute the bias. Please clarify*

The bias between the retrieval and the DSD measured by the disdrometers. We have clarified this in the revision.

*31. Figure 16:*

*a. In Figure 16b) also the R from original DSD ca be added*

The R derived from the original DSD is added in the graph, however the formatting of the graph makes this barely visible. We have replaced the format of the graph entirely because of legibility issues like these.

*b. please provide the label for x-axis*

We have replaced the format of the graph entirely, so this is no longer relevant.

***TECHNICAL CORRECTIONS***

*1. Section 2.2, first line: erase the word "second"*
*2. Figure 2: please put the legend outside the plot area, otherwise it covers some lines*
*3. Figure 5: Probably the Author can eliminate this figure and add the lamda-mu relation in Figure 4b. It is just a suggestion*
*4. Page 23, line n. 2: "Because" should be uppercase*

*5. Page 23, line n. 4: "It" should be lowercase*
*6. Section 7.2, first line: "Firstly" should be lowercase*

We thank the reviewer for pointing out these technical errors and have corrected them.

**Review 3**

**General comments:**

*The manuscript concerns estimation of drop size distribution (DSD) from attenuation of radiowaves at different frequencies/polarizations. The focal point of the manuscript lays in validation of proposed methods by numerical simulations, nevertheless applicability of the method is demonstrated also during a single rainfall event on attenuation measurements obtained from an experimental setup with dual polarized 38 GHz and horizontally polarized 26 GHz microwave links and array of disdrometers. The topic is relevant and the methodology is scientifically valid. The manuscript is also well structured and very well written.*

*My major concern is in applicability of the presented approach on real attenuation measurements obtained from commercial microwave links (CMLs), which is where the proposed methods have the highest potential. The DSD estimation is thoroughly tested on simulated attenuation observations, which are to my understanding ideal (not perturbed by any errors). This should be clearly stated probably already in the Method section because it is very important for interpretation of the results. The authors are apparently aware of different limits and pitfalls when it comes to application of the method on CML data, nevertheless the discussion of these limitations and pitfalls could (should) be more specific. This is important, because some of the conclusions based on numerical experiments (e.g. that dual frequency method is insensitive to difference between frequencies) are to my understanding only valid for ideal CMLs with high precision and accuracy. Detailed sensitivity or uncertainty analysis which would enable to quantify effect of inaccuracies in attenuation measurement on the efficiency of the proposed methods is probably out of the scope of this study, nevertheless the manuscript would clearly benefit from more robust discussion of the results in the context of real attenuation measurements from CML networks. This issue is further discussed in the specific comments.*

*Despite my concern regarding applicability of the method in real CML setting I consider the authors' work as a valuable contribution to DSD research and research related to exploitation of CMLs for environmental monitoring and believe that authors can address this issue by relatively minor revisions.*

We want to thank the reviewer for the compliments and encouragements and also for the due criticism. The reviewer points out that the applicability of the method may be limited in real CMLs and that the treatment of these limits should be more specifically addressed. In particular (as argued in the rest of the review) the potential inaccuracies resulting from wet antennas, quantization of the signal and the relative (in)sensitivity to rainfall at different frequencies and path lengths should be discussed in more detail. While we do not agree that all of these issues need to be necessarily addressed in this paper, we have expanded our analysis in particular with respect to the quantization error, which may be the most important limitation with regards to the practical application of this method. With respect to the inaccuracies due to wet antennas we believe that a thorough analysis of this issue would require too many assumptions to be of much value here

and the issue would be better investigated in a follow-up study.

**Specific comments:**

*P1L8 − 9 Abstract: Isn't the accuracy of the method highly dependent also on precision of the measurements as noted in the Conclusions or more general on accuracy of identified rainfall induced attenuation?*

Yes. We have extended this part of the abstract to reflect more of the conclusions in the revision.

*P3L10 − 14: Why do you use transect of 20 km when typical length of CMLs operated at frequencies 15 − 38 GHz (and esp. 26 − 38 GHz) is substantially shorter? Moreover, you later demonstrate the method on 2.2 km CML. Could you indicate the reason for simulating CML over entire length of the field?*

This was convenient because this was how the dataset was provided. The Length of the transect could be adapted although we do not look at the effect of link length in this paper.

*P7L2: The reasoning should probably refer to eq. 14 instead of Eq. 11 − 13 to apply not only for dual polarization setting but also for dual frequency setting. Furthermore, variables used in eq. 15 are defined in eq. 14 and not in Eqs. 11 − 13.*

Thanks for pointing out this inconsistency. This should indeed refer to eq. 14. We have adjusted this in the revision of our paper.

*P13 − 15: Please comment on spells with no results which can be seen on Fig. 10 (and later also on fig. 16). Are they due to not identified parameters, or due to measurement outages? Please, comment on these 'outages' also on P20.*

These gaps are due to the filtering described in section 2.2. We have added a note on this in the revised manuscript.

*P18L12 − P19L5 (the whole section): It is not clear if the offset is applied to both frequencies resp. polarizations. If yes, then the whole analysis is not much informative. And what if bias evolves in time differently for both polarizations? This might be the case e.g. due to wet antenna attenuation when droplets on the surface of antennas start to transform themselves into rivulets (see e.g. Mancini et al., 2019).*

The offset is applied to both frequencies and polarizations in equal measure. This is illustrative of wet antennas where the water forms a uniform layer.

*As noted in the Conclusion section, stability of baseline level is crucial in practice for utilizing the proposed technique. In practice, the feasibility of your DSD estimation approach will be probably very much sensitive to ratio between rainfall induced attenuation and other sources of attenua-*

tion which cannot be easily identified, i.e. it is reasonable to expect that method will be applicable only to CMLs relatively sensitive to rainfall (longer CMLs, higher frequencies) where effect of wet antenna attenuation and limited quantization is not so much pronounced. Given this, it might be much more informative to investigate sensitivity of the methods to precision of CMLs, e.g. simple averaging might very well simulate quantization of CMLs.

To analyse the effect of typical quantization strategies employed in operational CMLs to the simulated retrievals would be a valuable addition for this paper. We have added an extra subsection on this in the revised version.

I fully acknowledge that detail sensitivity or uncertainty analysis on the whole dataset might be sufficient for stand-alone work and it is out of the scope of this investigation, however, performing such analysis on a subset of data (e.g. the event used for demonstration on real data) would be probably sufficient to enable discussing applicability of the method in real CML network in more specific manner (see the next comment to the P23L1 − 11).

In any case, the limitations of recent sensitivity analysis should be addressed either in the section itself or within discussion section. It would be also valuable to i) refer to typical values of baseline offset and ii) to include into discussion of this issue also wet antenna effect. Finally consider presenting results in dB/km. This would make them applicable also to other link lengths.

A comprehensive treatment of the effect of wet antennas other than the case of a uniform layer of water is not feasible without additional experimental work, so we consider it out of scope for this paper. Presenting the offsets in dB/km is a good idea and we have adjusted the graph accordingly.

P23L1 − 11: The issues discussed in this paragraph are of crucial importance for application of the method in a real CML network. It would make sense to discuss in here also quantization of CMLs (which is raised at the end of the Conclusions without previous deeper discussion). Also discussion on wet antenna attenuation might be deeper, e.g. to which extend we can expect it will differ for two frequencies or polarizations? Finally, the results presented in the section Dependence on link frequency should be discussed in a view of inaccuracy of real CML measurements. It is likely that by lower frequencies (e.g. 17 GHz) only relatively long CMLs will have sufficient sensitivity to rainfall (and thus sufficient precision) to be suitable for DSD estimation. Similarly, in-sensitivity of DSD estimation on an offset between frequencies (now presented as one of the conclusions) does not apply for real data with limited precision. The attenuation difference will be for typical CML frequencies and lengths at least for light and moderate rainfalls below quantization of the CML records.

We have added an additional section on the effect of quantization and added further discussion pertaining to this issue.

P23L33 − P24L7: The second paragraph of the Caveats section lacks exactness of the previous text with vague formulations like 'potentially more serious', 'disdrometer measurements is used more directly', or 'somewhat in favor of'. Detailed investigation and quantitative assessment of discussed

*issues is probably out of the scope of this manuscript, nevertheless, authors might consider refor-mulating the second paragraph to provide more specific reasoning why and to which extend can the mask influence the results and overall applicability of the proposed methods.*

We have reformulated this paragraph to be more specific.

*P23L29: typo, 'too' instead of 'to'*

Thanks for pointing out the typo.

*P25L15 − 17: The limited precision of CMLs due to quantization is a very important limitation which should be discussed in more detail already in the Discussion section(see comment to P23L1 −11).*

We have expanded on the effects of quantization in the revision.

*Some further questions I would be curious about (no need to answer):*
*- The three parameter method which uses as an input three attenuation measurements(k1,2,3) provided less accurate DSD estimate than two-parameter method. Have you tried to apply two-parameter method on different combinations of attenuations (k1/k2,k1/k3,k2/k3) and use the re-dundant parameter estimates for improving the estimation accuracy (e.g. identifying outliers)?*

We have not done this. This is an interesting suggestion that we might pursue in a follow-up paper.

*- There is probably some autocorrelation in parameters of DSD function, have you thought about using autocorrelation structure of DSD observations to constrain the optimization procedure?*

We have not looked into the autocorrelation structure specifically. This is an interesting suggestion.

**General reaction**

We want to thank all three reviewers for their useful feedback. To summarize, we have identified the following major points that we feel confident have been addressed in the revised manuscript:

- Include analysis and discussion regarding the effect of quantization on the retrieval accuracy.
- Replace the time-series figures with a more informative format.
- Provide failure rates (no convergence) for the different retrieval variants.

We do not believe that expanding further on the experimental results fits in the scope of the paper, nor do we believe that further analysis based on the current experimental data would yield major improvements. The principal scope of the paper is with the simulated retrievals. At the discretion of the editor we will consider removing the experimental part entirely or be more explicit about the tentative nature of the experimental part. Even so, we prefer not to do the former.
We have of course addressed all relevant minor technical errors.

**List of changes**

Changes in the text, all page and line numbers refer to the new text:

- Abstract P1L8-10 (Extended text)

- Section 1, P1L15-16 (added reference and mention of satellite uplinks)
- Section 1, P1L17 (minor punctuation)
- Section 1, P2L4 (minor textual)
- Section 1, P2L5 (minor spelling and punctuation)
- Section 1, P2L16 (minor punctuation)

- Section 2.1, P3L4-6 (Clarification and fixed reference)
- Section 2.1 P3L8-10 (added information on the events)
- Section 2.1 P3L12-13 (Added explicit reference)
- Section 2.2 P4L2 (replace "dD" with "diameter")
- Section 2.2 P4L15 (minor textual)

- Section 3.1 P5L18 (added "corresponding")
- Section 3.1 P6L9 (minor punctuation)
- Section 3.2 P7L4 (fixed equation symbol)
- Section 3.2 P7L6 (fixed equation number)
- Section 3.2 P7L7 (fixed equation symbols)
- Section 3.2 P7L12-13 (reformulated)
- Section 3.2 P7L14-15 (fixed symbols)
- Section 3.2 P7L16 (Added reference)
- Section 3.2 P7L17 (minor reformulation)
- Section 3.2P7L23 (fixed symbol)
- Section 3.2 P7L26 (added acronym)

- P9 Figure 4 (Caption reformulated and extended)

- Section 3.3 P9L2 (added reference)
- Section 3.3 P9L8 (minor punctuation)
- Section 3.3 P9L10 (fixed symbols)
- Section 3.4 P9L15 (minor style fixes)
- Section 3.4 P10L4-8 (Expanded paragraph and removed reference)
- Section 3.4 P10L18-20 (added description of 'failure ratio')

- Section 4.1 P11L4 (fixed frequencies mentioned)
- Section 4.1 P11L12 (changed "partial" to "specific")
- Section 4.1 P11L14 (changed "partial" to "specific")
- Section 4.2 P11L25 (figure number)

- Section 4.2 P11L26 (changed time notation)
- Section 4.2 P12L1 (acronym)
- Section 4.2 P12L2 (reformulated)
- Section 4.2 P12L4 (changed time notation)
- Section 4.2 P12L6 (addition)
- Section 4.2 P12L7 (figure number)
- Section 4.2 P12L8 (acronym)
- Section 4.2 P12L10-11 (reformulated)

- P13 Figure 7 (replaced caption)
- P14 Figure 8 (replaced caption)

- Section 5 P15L8-11 (description corrected and expanded)
- Section 5.1 P15L19 (Clarified)

- P16 Figure 10 (replaced caption)
- P17 Table 2 (expanded caption and minor style fixes)
- P18 Table 3 (expanded caption)
- P19 Figure 10 (expanded caption for extra panel)

- Section 5.1 P17L10-11 (figure numbers, addition)
- Section 5.1 P17L12 (added "actually")
- Section 5.1 P17L13 (reformulated)
- Section 5.2 P18L6-7 (clarified)
- Section 5.3 P20L5 (clarified)
- Section 5.4 P20L14-P21L13 (new section)

- Section 6 P21L17 (minor punctuation)
- Section 6 P22L1 (minor capitalization)
- Section 6 P22L2-4 (added text)
- Section 6 P22L6 (minor punctuation)
- Section 6 P22L7-8 (table numbers, removed text)
- Section 6 P22L9 (minor grammar)
- Section 6 P22L11 (clarification)
- Section 6 P22L13 (minor punctuation)
- Section 6 P22L15-P23L1 (added text)
- Section 6 P23L1-2 (reformulated)

- P24 Figure 15 (replaced caption)

- Section 7.1 P23L5 (minor reformulation)
- Section 7.1 P23L8 (minor capitalization)
- Section 7.1 P25L9-14 (added text)
- Section 7.2 P26L19-21 (reformulated)

- Section 7.2 P26L23-28 (reformulated)

- Section 8 P27L9 (added text)
- Section 8 P27L10 (minor grammar)
- Secton 8 P27L11 (addition)
- Secton 8 P27L17 (minor grammar)
- Section 8 P27L18 (clarification)
- Section 8 P27L19 (deleted text)
- Section 8 P27L21 (changed 'good' to 'reasonable')
- Section 8 P27L22 (addition)
- Section 8 P27L23 (minor reformulation)
- Section 8 P27L25-26 (addition)
- Section 8 P28L5 (minor reformulation)
- Section 8 P28L6 (added 'severely')

Changes in figures and tables, all figure and table numbers refer to the new text:

- Figure 4 (replaced panel b, consequently removed the old figure 5)
- Figure 6 (reduced number of lines, reordered line colors, added different line styles)
- Figure 7 (completely replaced)
- Figure 8 (completely replaced)
- Figure 9 (completely replaced)
- Table 2 (added failure ratios, corrections to calculations)
- Table 3 (added failure rations, corrections to calculations)
- Figure 10 (logarithmic axis, additional panel)
- Figure 11 (Slight layout changes)
- Table 4 (new)
- Table 5 (added failure ratios, corrections to calculations)
- Figure 14 (improved axis labels)
- Figure 15 (completely replaced)

---

## Referee Report (RR1)

**Estimating raindrop size distributions using microwave link measurements**
**by T. van Leth, H. Leijnse, A. Overeem and R. Uijlenhoet**

**Assessment:**
This is the second time I review this paper and my overall impression of it remains rather negative. I'm particularly disappointed in the way the authors handled my previous comments. Some superficial changes were made but the really important issues regarding feasibility and validation remain the same (see below for more details). In their rebuttal, the authors say that "the suggestion to dig deeper into the experimental data is not helpful and further analysis over the whole experimental dataset would not yield meaningful results." I don't agree with this assesment and encourage the authors to reconsider their position. In particular, I don't think that there is enough scientific evidence to support the feasibility of the retrieval methods yet. The simulation study is interesting but highly idealized and far away from reality. Given that there is still no proper uncertainty/error analysis, it is hard to judge the soundness of the retrievals. Unfortunately, since the authors do not seem to be interested in performing more detailed and rigorous investigations, I cannot recommend publication at this point.

**Main arguments against publication:**
1. There is no rigorous and realistic assessment of the uncertainty affecting the retrieved DSD parameters (e.g., no error bars and no benchmark for comparing results). The simulation studies are performed in idealized conditions which do not reflect reality.

2. The presented evidence does not always match the conclusions/statements made by the authors. There are several inconsistent and contradicting sentences (see below). The general conclusion of the paper regarding feasibility remains unclear.

3. The writing is biased towards highlighting potential rather than providing a fair objective scientific assessment of feasibility and accuracy.

**A. Feasibility**

**A1.** The authors base most of their conclusions on a few, highly idealized simulation studies. But to me, these are of little practical and scientific value. In reality, there are serious issues due to the instability of the baseline, quantization and wet antenna attenuation which make the proposed techniques very unlikely to be ever applicable to commercial microwave networks. Indeed, Figure 5 shows that the relationship between the attenuation ratio and the value of mu is almost flat. To get a good accuracy on mu, one therefore needs a very high accuracy on the attenuation ratio. *"To achieve a non-convergence ratio of 10%, quantization errors of 0.001 dB would be required."* However, current accuracies are 0.1 dB at best, which is several orders of magnitude lower than what is actually required. Higher accuracies are unlikely to be ever available in commercial networks due to the high cost of measuring power more accurately and other technical limitations (e.g., additional uncertainty due to baseline and wet antenna).

**A2.** Figure 14b clearly shows that the attenuation ratios derived from actual data are extremely noisy and poorly correlated with the true attenuation ratios. And this is for a "good" case without any quantization noise. Sure, you can cherry pick a few decent retrievals in there. But these might as well be coincidences and there is not enough hard evidence to prove feasibility. Please consider more cases and/or perform a more systematic and rigorous assessment.

**B. Validation and assessment:**

**B1.** The approach used to validate the DSD retrievals using MOR, MAD and AD95 on N(D) and R is inadequate. At best, it's incomplete. On their own, these values don't mean anything! A proper validation requires a benchmark against which the reported performances can be compared. For example, if the goal is to retrieve the rainfall rate from the links, then you should validate against the alternative model of retrieving R through the power-law relation $A = aR^b$ (without any knowledge of the DSD). If your method does not perform better than that, then there is no skill in the retrieved DSDs for the rainfall estimation problem. Similarly, if your goal is to retrieve the Dm or mu values, then you should validate against the alternative model which assumes a constant value (e.g., the climatological mean). In any case, error bars and a rough estimate of the uncertainty affecting the retrieved quantities need to be provided!

**B2.** It is not 100% clear how MOR, MAD and 95AD were calculated. Please provide unambiguous expressions/equations for your performance scores and clarify the difference between the "normalized" and non-normalized versions.

**B3.** It would be good to show a few cases in which the retrievals failed in order to have a better understanding of the numerical issues involved and the type of measurements that cause the algorithm(s) to fail. Right now, the paper mostly focuses in highlighting good cases, which is only one side of the story.

**B4.** The sensitivity study in 5.3 is based on unrealistic assumptions. The use of an equal offset for both frequencies/polarizations is much too optimistic. In reality, the errors/offsets on the individual measurements are likely to be independent. Indeed, the final offset is the result of many error/noise terms from multiple factors such as electronics, baseline attenuation, wet antenna and quantization effects. By assuming the same offset for both measurements, you are dramatically underestimating the uncertainty affecting the attenuation ratios. Please use independent offsets during the sensitivity study or justify why you think it is appropriate to use correlated noise terms.

**B5.** Page 18, ll.16-17: Why don't you take the effect of noise into account in the simulations. Please explain!

**B6.** Page 13, Figure 7: There are important conditional biases in the retrievals of Nt and mu. But almost no explanations are given to what caused them. A more detailed discussion is needed to understand these results and how they affect the quality of the DSD retrievals.

**C. Inconsistent and/or misleading statements:**

**C1.** Page 1 (abstract): *"Simulations show that a DSD retrieval on the basis of microwave links can be highly accurate."* This is a strong statement that is not aligned with the evidence presented in the paper. In reality, the simulations show that even under idealized conditions, the retrievals can fail. Please reformulate.

**C2.** Page 25, ll. 8-9 the authors write that: *"This provides a hopeful perspective for the application to commercial networks."* However, this is not really consistent with the other statements made in the paper. For example, on page 21, l.7 it is said that *"This limits the prospective of successful application to current networks"*. On page 25, ll. 5-6, *"the links examined in this study lacked stability […] and wet antenna attenuation was an intractable problem"*. On page 25, ll. 11-13, *"The only way to apply such retrievals to currently operational unmodified link networks consistently is to install dedicated data-loggers at selected link locations to read out the analog signal directly, which might not be feasible."*

**C3.** Page 25, ll.2-3: *"No such problem exists in principle with regard to the phase difference; it is independent of any baseline as long as that baseline is indifferent to polarization."* Yes, but there is no

evidence that the baseline is actually indifferent to polarization and wet antenna attenuation. Please reformulate the sentence to avoid misunderstandings.

**C4.** Page 27 (conclusions): *"... we have shown that a DSD retrieval on the basis of multiple microwave link variables can be successful and highly accurate, but only when precise high-resolution records of received power are available."* This is a very misleading statement. Firstly, the conditions under which a retrieval can be made and the uncertainty affecting the retrieved values remain unclear (i.e., due to the lack of a proper uncertainty analysis). Secondly, what is really needed for a successful retrieval is a precise measurement of the "rain-induced attenuation" and not the "received power". That's a big difference because in practice, it is almost impossible to get a precise rain-induced attenuation estimate, even if you could measure the received power accurately. Please reformulate to convey the right meaning.

**D. Others:**

**D1.** The discussion about computation time is not really relevant. There are no real challenges associated with the numerical optimization techniques used in this study and real-time implementation would not be a problem. I suggest to shorten this part or remove it in favor of a more detailed uncertainty assessment.

**D2.** Page 4, section 2.2: The temporal resolution of the DSD data are missing. I assume it's 30s?

**D3**. Page 8, l.13: the figure number is missing

**D4.** Page 10, l.17: What do you mean by "real outliers?" As opposed to imaginary ones?

**D5.** Page 12, ll.1-2 *"These outliers do not seem to correspond with any … with high drop concentrations". Why? Can you elaborate?*

**D6.** Page 12, l.10: of the rain intensity  are given in Table 1

**D7.** Page 19, Figure 11: Please provide units for MOR, MAD and 95AD and specify what quantity is considered here (N(D) or N(D)/Nt?).

**D8.** Page 21, l.12: "This  can be attributed"

**D9.** Page 26, the threshold used to select DSDs for inferring the mu-lambda relationship is not what I call a "compromise". It's a fixed threshold imposed by the authors based on a previous paper without any justification or optimization. Please reformulate.

**D10.** Page 26: "Considering the small spatial scale of the measurements we considered and the high spatial correlations therein this is an acceptable loss". This sentence does not make any sense.

**D11.** Page 26-27: The discussion about the truncation of the gamma distribution on page is besides the point. The real issue is not the truncation but the fact that real DSDs are never perfectly gamma. Even if they were distributed according to a gamma at the point scale, the average DSD along the link path would be a mixture of gamma distributions with different shape parameters which is not a gamma anymore. To me, the whole discussion about the truncation issue seems to be a minor issue in this story. Instead of obsessing about it, the authors could provide more details about the sampling uncertainty affecting the retrieved DSD estimates or the sensitivity to the temporal resolution of the link data.

**D12.** Page 27: "… but the effects of this on high order moments is minute". Does not make any sense. Please reformulate.

---

## Referee Report (RR2)

**REVIEW REPORT**

Review of amt-2019-51-manuscript-version4

By Thomas C. van Leth, Hidde Leijnse, Aart Overeem, and Remko Uijlenhoet

Manuscript Title – Estimating raindrop size distributions using microwave link measurements

**MINOR COMMENTS**

In my opinion the Authors have addressed all my concerns/question improving the quality of the manuscript. I suggest minor revision before the publication. The main revision regard several sentence within the text that probably refer to the figures reported in the previous version of the manuscript and therefore describe something that is not shown in the referred Figure. It can be solved modifying the sentence or adding *(not shown)* in the text. The latter happen at:

- Page 12 3$^{rd}$ line: "The temporal evolution…."
- Page 12 last 3 lines: "The retrieval gives an …."
- Page 15 "Furthermore, fort both the retrieval methods,…."
- Page 22 "The resulting DSD is very similar…."
- Page 27 1$^{st}$ line: the list of the Figure is wrong and all the sentences until the end of the Section need to be checked.

Please check if there are other sentences in similar conditions that I did not notice within all the manuscript.

Below some few suggestion:

- Page 8 last paragraph: "….(as shown in Fig??)…."
- Page 10, line n. 5: please quantify "significantly". Which is the allowed differences between retrieved and TS96 parameters to consider that the gamma distribution is not a good fit?
- Page 10, line 6: please quantify "significantly"
- Page 10, lines 7-8: Which is the number of time that the measured DSD deviate significantly from the TS96 DSD? Which is the number of time that the estimated DSD deviate significantly from the measured DSD? Please add this information
- Table 2 (and all the tables that report the same statistics) : For sake of simplicity please add the unit. fail is a percentage?

---

## Author Response (AR2)

**REBUTTAL**

**amt-2019-51, Submitted on 08 Feb 2019**

**Estimating raindrop size distributions using microwave link measurements**

Thomas C. van Leth, Hidde Leijnse, Aart Overeem, and Remko Uijlenhoet

**Associate Editor Decision: Publish subject to technical corrections (25 Jan 2020) by Saverio Mori**

Comments to the Author:

Dear Authors, In my opinion this work can be published; nevertheless several important corrections are absolutely required. Anonymous referees have given precise indications in this respect, both on minor comments and on major and substantial ones. Without an adequate addressing of the issues indicated, this work is not suitable for publication.

Dear editor, thank you very much for communicating your decision. We have addressed all issues indicated. See our detailed replies to the referees' comments and suggestions, as well as our revised manuscript (with adjustments with respect to the previous version indicated in red).

Non-public comments to the Author:

Dear Authors, first i apologize for the delay within the publication process. I have had several doubts on how proceeding, because of the analysis mine and of the anonymous referees. I have decided to proceed, nevertheless i condivide the most critical comments of referees #1 and #3, and you should adequately address them. The proposed approach is interesting but based on idealistic assumptions: this must be clear in all the paper, and the assumptions clearly indicated and described; also conclusions have to be corrected. All The Best

Dear editor, we apologize for the delay in our response. This was due to personal circumstance beyond our control. That said, in the revised version of our paper we have followed your recommendation to make it even more clear to the reader that the adopted approach represents conditions where measurement errors and uncertainties associated with microwave link measurements do not play a role. We would like to stress, however, that we did test the proposed methodology on measurements from actual microwave link instruments during one event (Section 6). In any case, our assumption of idealized conditions has now been articulated even better throughout our paper, although we would like to recall that we already discussed the practical limitations of our method in the previous version of our paper (e.g. in the last sentence of the abstract and quite extensively in the discussion section). To avoid any misconception on the part of the reader, we have extended the title of our paper to "Estimating raindrop size distributions using microwave link measurements: potential and limitations".

**Assessment:**

This is the second time I review this paper and my overall impression of it remains rather negative. I'm particularly disappointed in the way the authors handled my previous comments. Some superficial changes were made but the really important issues regarding feasibility and validation remain the same (see below for more details). In their rebuttal, the authors say that "the suggestion to dig deeper into the experimental data is not helpful and further analysis over the whole experimental dataset would not yield meaningful results." I don't agree with this assessment and encourage the authors to reconsider their position. In particular, I don't think that there is enough scientific evidence to support the feasibility of the retrieval methods yet. The simulation study is interesting but highly idealized and far away from reality. Given that there is still no proper uncertainty/error analysis, it is hard to judge the soundness of the retrievals. Unfortunately, since the authors do not seem to be interested in performing more detailed and rigorous investigations, I cannot recommend publication at this point.

We respectfully disagree with this referee. We believe that our previous revision included significantly more than "some superficial changes". We would like to stress that our paper is certainly not meant to be the last word on estimating raindrop size distributions using microwave link measurements. Rather, it should be seen as an extensive feasibility study, both under simulated conditions (using simulated DSD fields derived from polarimetric radar observations for two rainfall events) and real conditions (using a ninemonth dataset from a line configuration of five laser disdrometers). We also tested the proposed methods on measurements from actual microwave link instruments during one event along the same path as the disdrometers. Finally, we sincerely believe that our paper does involve "detailed and rigorous investigations", although we admit that a full error analysis of the proposed DSD retrieval methods is beyond the scope of the current paper. We aim to address this in future work.

**Main arguments against publication:**

1. There is no rigorous and realistic assessment of the uncertainty affecting the retrieved DSD parameters (e.g., no error bars and no benchmark for comparing results). The simulation studies are performed in idealized conditions which do not reflect reality.

Although we have not included a full error analysis (see our response above), we do present a discussion of the sensitivity to attenuation bias (Section 5.3) and power quantization error (Section 5.4). As mentioned above, a full error analysis of the proposed DSD retrieval methods is beyond the scope of the current paper. Note that the simulation studies are based on real radar or disdrometer data, where the latter span a period of nine months representative for rainfall in the Dutch climate.

2. The presented evidence does not always match the conclusions/statements made by the authors. There are several inconsistent and contradicting sentences (see below). The general conclusion of the paper regarding feasibility remains unclear.

We thank the reviewer for identifying occasions where our statements could be interpreted as favoring the potential rather than the limitations of the proposed methodology. We have rephrased these statements wherever relevant. Moreover, we changed the title to better reflect the limitations.

3. The writing is biased towards highlighting potential rather than providing a fair objective scientific assessment of feasibility and accuracy.

See previous response.

**A. Feasibility**

**A1.** The authors base most of their conclusions on a few, highly idealized simulation studies. But to me, these are of little practical and scientific value. In reality, there are serious issues due to the instability of the baseline, quantization and wet antenna attenuation which make the proposed techniques very unlikely to be ever applicable to commercial microwave networks. Indeed, Figure 5 shows that the relationship between the attenuation ratio and the value of mu is almost flat. To get a good accuracy on mu, one therefore needs a very high accuracy on the attenuation ratio. "*To achieve a non-convergence ratio of 10%, quantization errors of 0.001 dB would be required."* However, current accuracies are 0.1 dB at best, which is several orders of magnitude lower than what is actually required. Higher accuracies are unlikely

to be ever available in commercial networks due to the high cost of measuring power more accurately and other technical limitations (e.g., additional uncertainty due to baseline and wet antenna).

We respectfully disagree with the referee that we "base most of [our] conclusions on a few, highly idealized simulation studies". As mentioned above, we test the feasibility of the proposed DSD retrieval methods both under simulated conditions (using simulated DSD fields derived from polarimetric radar observations for two rainfall events), real conditions (using a nine-month dataset from a line configuration of five laser disdrometers) as well as measurements from actual microwave link instruments during one rainfall event along the same path as the disdrometers. That is significantly more than "a few, highly idealized simulation studies". Indeed, it will be a challenge applying the proposed methods to commercial microwave link networks. But one first needs to perform a feasibility study such as the one we have performed to learn that such is actually the case. In addition, we believe that reporting less favorable results is also important for scientific progress. Even then, our methodology may be applicable to dedicated research microwave link configurations such as the one described in Section 2.2 of our paper. Such a setup may then be seen as a (very) large disdrometer.

**A2.** Figure 14b clearly shows that the attenuation ratios derived from actual data are extremely noisy and poorly correlated with the true attenuation ratios. And this is for a "good" case without any quantization noise. Sure, you can cherry pick a few decent retrievals in there. But these might as well be coincidences and there is not enough hard evidence to prove feasibility. Please consider more cases and/or perform a more systematic and rigorous assessment.

As was indicated in the first sentences of Section 6 of our paper: "The baseline power level of the links showed considerable fluctuations over the course of the measurement period. Therefore, it was not feasible to perform retrievals for the entire 9-month dataset". This is an unfortunate situation which we are, alas, not able to revoke at this stage. The measurement campaign with the microwave link setup took place between 1 April 2015 and 1 January 2016. The experimental setup has since been disassembled. Actually, the building where one end of the microwave link setup was installed no longer exists. To accuse us of "cherry picking" at this stage does not feel fair. We believe we have made a serious attempt to demonstrate the challenges of applying the proposed DSD retrieval methods to actual microwave link measurements. Furthermore, we have selected the event that we present based on a relatively stable baseline around the event, rather than on the performance of the retrieval, as was clearly stated on p.21, line 18. An assessment on a more extensive dataset would mean setting up a new microwave link measurement campaign. That is certainly beyond the scope of the current work.

**B. Validation and assessment:**

**B1.** The approach used to validate the DSD retrievals using MOR, MAD and AD95 on N(D) and R is inadequate. At best, it's incomplete. On their own, these values don't mean anything! A proper validation requires a benchmark against which the reported performances can be compared. For example, if the goal is to retrieve the rainfall rate from the links, then you should validate against the alternative model of retrieving R through the power-law relation  $A = aR^b$  (without any knowledge of the DSD). If your method does not perform better than that, then there is no skill in the retrieved DSDs for the rainfall estimation problem. Similarly, if your goal is to retrieve the Dm or mu values, then you should validate against the alternative model which assumes a constant value (e.g., the climatological mean). In any case, error bars and a rough estimate of the uncertainty affecting the retrieved quantities need to be provided!

As the title of our paper indicates, our aim was not to retrieve R alone, but rather to retrieve DSD parameters. The simulation framework we employed, based on two events with spatial DSD fields derived from polarimetric weather radar and nine months of disdrometer data, allowed us to explore the feasibility of DSD retrieval in a controlled environment, i.e. under conditions where one knows the true DSD (and hence the true R). In such a situation, the reference is not the climatological mean but the true value of a DSD parameter. Therefore, we think that the validation approach is adequate given the goal of this paper. Finally, concerning the actual microwave link measurements discussed in Section 6, we have not included a full error analysis, but – as we mentioned above – we do present a discussion of the sensitivity to attenuation bias (Section 5.3) and power quantization error (Section 5.4). Once more, a full error analysis of the proposed DSD retrieval methods is beyond the scope of the current paper.

**B2.** It is not 100% clear how MOR, MAD and 95AD were calculated. Please provide unambiguous expressions/equations for your performance scores and clarify the difference between the "normalized" and non-normalized versions.

We believe the definitions of MOR, MAD and 95AD provided in Section 3.4 represent unambiguous descriptions of our performance measures. As stated in the same section, "all metrics are normalized with respect to the median of the original quantities". We added ", hence they are dimensionless" to this sentence and "(also dimensionless)" after "the failure ratio" to further clarify the employed metrics. We believe that should be clear to the reader. In addition, none of the other referees asks for "expressions/equations" of the statistical measures we use.

**B3.** It would be good to show a few cases in which the retrievals failed in order to have a better understanding of the numerical issues involved and the type of measurements that cause the algorithm(s) to fail. Right now, the paper mostly focuses in highlighting good cases, which is only one side of the story.

We state toward the end of Section 3.4 that "we also compute the fraction of non-convergent retrievals compared to the total number of retrievals. This 'failure ratio' is necessary for a complete picture of the robustness of the method since the other metrics naturally exclude these intervals". Tables 2 and 4 report the values of this failure ratio for rainfall retrievals based on microwave link simulations for the nine-month disdrometer dataset. Finally, Section 6 provides an application of the proposed retrieval methods to actual microwave link measurements for a complete rainfall event, clearly showing the challenges and limitations of the proposed methods. Hence, in all honesty, we believe we pay ample attention to "the numerical issues involved and the type of measurements that cause the algorithm(s) to fail".

**B4.** The sensitivity study in 5.3 is based on unrealistic assumptions. The use of an equal offset for both frequencies/polarizations is much too optimistic. In reality, the errors/offsets on the individual measurements are likely to be independent. Indeed, the final offset is the result of many error/noise terms from multiple factors such as electronics, baseline attenuation, wet antenna and quantization effects. By assuming the same offset for both measurements, you are dramatically underestimating the uncertainty affecting the attenuation ratios. Please use independent offsets during the sensitivity study or justify why you think it is appropriate to use correlated noise terms.

We respectfully disagree with the reviewer that the offsets are likely to be independent, as baseline fluctuations and wet antenna attenuation will affect multiple links in a similar manner. Of course, we realize that the assumption of an (equal) offset for all attenuations is hardly ever completely met in practice. However, it provides a reasonable first order appreciation of the effects of attenuation biases, which was the purpose of the analysis presented in Section 5.3.

**B5.** Page 18, II.16-17: Why don't you take the effect of noise into account in the simulations. Please explain!

Because that is beyond the scope of the current paper. As stated, "it is expected that this would influence the retrieval the most when the frequencies are close together". We have decided to study the effect of offsets because this is known to be the largest source of error in microwave link rainfall monitoring, and it is hence expected that noise would have a smaller effect than offsets. It would certainly be very interesting and relevant to study this effect in detail, but that would effectively lead to an additional paper. We are motivated to address this issue in future work.

**B6.** Page 13, Figure 7: There are important conditional biases in the retrievals of Nt and mu. But almost no explanations are given to what caused them. A more detailed discussion is needed to understand these results and how they affect the quality of the DSD retrievals.

We agree that it would be interesting to learn more about the reasons for the mentioned biases, but unfortunately we have no clear explanation for them. As mentioned in Section 4.2 "These outliers do not seem to correspond with any particularly high or low precipitation intensity, but they do correspond with high drop concentrations". There is not much more we can meaningfully say about this.

**C. Inconsistent and/or misleading statements:**

**C1.** Page 1 (abstract): "*Simulations show that a DSD retrieval on the basis of microwave links can be highly accurate."* This is a strong statement that is not aligned with the evidence presented in the paper. In reality, the simulations show that even under idealized conditions, the retrievals can fail. Please reformulate.

We have adapted the formulation in the revised paper by removing "highly" and adding "under idealized conditions" at the end of this sentence. Note that we already stated in the last sentence of the abstract

that "in practice, the accuracy and success rate of any retrieval is highly dependent on the stability of the base power level as well as the precision of the instruments and in particular the quantization applied to the recorded power level".

**C2.** Page 25, II. 8-9 the authors write that: "*This provides a hopeful perspective for the application to commercial networks.*" However, this is not really consistent with the other statements made in the paper. For example, on page 21, I.7 it is said that "*This limits the prospective of successful application to current networks*". On page 25, II. 5-6, "*the links examined in this study lacked stability* [...] and wet antenna attenuation was an intractable problem". On page 25, II. 11-13, "*The only way to apply such retrievals to currently operational unmodified link networks consistently is to install dedicated data-loggers at selected link locations to read out the analog signal directly, which might not be feasible."*

That particular statement ("This provides a hopeful perspective for the application to commercial networks") refers to the sentence before, which reads: "we found that a former commercial microwave link had a much stabler baseline and furthermore the effect of wet antennas was much more manageable for that particular device". In other words, this was not a general qualification of the potential of commercial microwave links for DSD retrieval, but rather a statement about the (perhaps surprising) stability of the baseline of a former commercial microwave link as compared to a dedicated research link. To clarify this further, we added at the end of the mentioned statement: ", in particular if data could be logged with high precision (Chwala et al., 2012)". Apart from this, we do not see a good reason to rephrase this statement.

**C3.** Page 25, II.2-3: "*No such problem exists in principle with regard to the phase difference; it is independent of any baseline as long as that baseline is indifferent to polarization."* Yes, but there is no evidence that the baseline is actually indifferent to polarization and wet antenna attenuation. Please reformulate the sentence to avoid misunderstandings.

We thank the reviewer for pointing this out. The condition that the baseline is indifferent to polarization is not necessary. We meant that the base line issues which affect rainfall retrievals based on measurements of signal amplitude do not play a role for retrievals based on signal phase differences between orthogonal polarizations. Wet antenna attenuation will affect the received signal amplitude, but not the phase difference at the receiver. We shortened this sentence to: "[...]; it is independent of any power baseline".

**C4.** Page 27 (conclusions): "... we have shown that a DSD retrieval on the basis of multiple microwave link variables can be successful and highly accurate, but only when precise high-resolution records of received power are available." This is a very misleading statement. Firstly, the conditions under which a retrieval can be made and the uncertainty affecting the retrieved values remain unclear (i.e., due to the lack of a proper uncertainty analysis). Secondly, what is really needed for a successful retrieval is a precise measurement of the "rain-induced attenuation" and not the "received power". That's a big difference because in practice, it is almost impossible to get a precise rain-induced attenuation estimate, even if you could measure the received power accurately. Please reformulate to convey the right meaning.

You are right. We have removed "highly" and we have replaced "received power" by "rain-induced attenuation".

**D. Others:**

**D1.** The discussion about computation time is not really relevant. There are no real challenges associated with the numerical optimization techniques used in this study and real-time implementation would not be a problem. I suggest to shorten this part or remove it in favor of a more detailed uncertainty assessment.

This "discussion" is actually only one paragraph in Section 7.1. Because the computational burden is an important distinction between the two-parameter and the three-parameter retrieval method, we think it is relevant for the reader to know about this. In fact, this distinction may even become more relevant in an operational setting, where real-time computations are required, especially if this is done on embedded link hardware (Chwala et al., 2012).

D2. Page 4, section 2.2: The temporal resolution of the DSD data are missing. I assume it's 30s?

Indeed, thanks. We added this in Section 2.2, also for the microwave links.

D3. Page 8, I.13: the figure number is missing

Thanks. That should have been Fig. 4b. This is now corrected.

**D4.** Page 10, I.17: What do you mean by "real outliers?" As opposed to imaginary ones?

Haha. We meant the actual "outliers", which fall outside the core of the error distribution. We replaced "real outliers" by "true outliers".

**D5.** Page 12, II.1-2 "These outliers do not seem to correspond with any ... with high drop concentrations". Why? Can you elaborate?

Unfortunately, no. As indicated above, we would have liked to understand this issue ourselves, but have not succeeded in doing so.

D6. Page 12, I.10: of the rain intensity is are given in Table 1

Thanks. "is given in Table 1" has been replaced by "are given in Table 1".

**D7.** Page 19, Figure 11: Please provide units for MOR, MAD and 95AD and specify what quantity is considered here (N(D) or N(D)/Nt?).

As indicated in the caption, "all statistics are normalized with respect to the median of the moment of the original measured DSD". Hence, MOR, MAD and 95AD are dimensionless in Fig. 11. This has now also been explicitly stated in Section 3.4. Also, as indicated in the caption, what is considered here is "the third order moment of the DSD [...] as a function of carrier frequency". Hence, this is neither N(D) nor N(D)/Nt.

D8. Page 21, I.12: "This is can be attributed"

Thanks. We removed "is".

**D9.** Page 26, the threshold used to select DSDs for inferring the mu-lambda relationship is not what I call a "compromise". It's a fixed threshold imposed by the authors based on a previous paper without any justification or optimization. Please reformulate.

It is a "compromise" between the desire to be comprehensive (take all measured DSDs into account no matter how small the sample size) and the desire to be selective (only take those DSDs into account that correspond to significant sample sizes and hence rain rates). One could call this threshold an "educated guess", but we believe "compromise" actually reflects best what we mean. Therefore, we have decided to keep it.

**D10.** Page 26: "Considering the small spatial scale of the measurements we considered and the high spatial correlations therein this is an acceptable loss". This sentence does not make any sense.

We replaced this sentence with: "Considering that the employed disdrometers were located relatively close together and therefore that their measurements are strongly correlated, we accepted this potential bias".

**D11.** Page 26-27: The discussion about the truncation of the gamma distribution on page is besides the point. The real issue is not the truncation but the fact that real DSDs are never perfectly gamma. Even if they were distributed according to a gamma at the point scale, the average DSD along the link path would be a mixture of gamma distributions with different shape parameters which is not a gamma anymore. To me, the whole discussion about the truncation issue seems to be a minor issue in this story. Instead of obsessing about it, the authors could provide more details about the sampling uncertainty affecting the retrieved DSD estimates or the sensitivity to the temporal resolution of the link data.

We are not "obsessed" with truncation effects; we simply discuss it as one of the caveats (Section 7.2) of our approach. Nothing more and nothing less.

**D12.** Page 27: "... but the effects of this on high order moments is minute". Does not make any sense. Please reformulate.

We meant "minute" in the sense of "(very) small", "minimal" or "marginal".

**REVIEW REPORT**

Review of amt-2019-51-manuscript-version4

By Thomas C. van Leth, Hidde Leijnse, Aart Overeem, and Remko Uijlenhoet

Manuscript Title - Estimating raindrop size distributions using microwave link measurements

MINOR COMMENTS

In my opinion the Authors have addressed all my concerns/question improving the quality of the manuscript. I suggest minor revision before the publication. The main revision regard several sentence within the text that probably refer to the figures reported in the previous version of the manuscript and therefore describe something that is not shown in the referred Figure. It can be solved modifying the sentence or adding *(not shown)* in the text.

We thank the referee for his/her positive evaluation of our revised paper as well as for the suggestions for further improvement. We have taken all of them into account in the revised version of our paper (see below for our detailed responses).

The latter happen at:

- Page 12 3rd line: "The temporal evolution...."

We added "(not shown)" to the end of the sentence "The temporal evolution of  $\mu$  is very close to the temporal evolution of  $\Lambda$ , with a correlation coefficient of 0.86."

- Page 12 last 3 lines: "The retrieval gives an ...."

We added "(not shown)" after "The retrieval".

- Page 15 "Furthermore, fort both the retrieval methods,...."

We added "(not shown)" after "the retrieval".

- Page 22 "The resulting DSD is very similar...."

We added "(not shown)" to the end of the sentence "The resulting DSD is very similar in shape to that obtained in the simulations, with overestimations especially at smaller diameters, but with the general shape of the DSD preserved".

- Page 27 1st line: the list of the Figure is wrong and all the sentences until the end of the Section need to be checked.

We replaced "Figs. 6a, 8b and d, 9c, 10 and 15c" with "Figs. 6a, 10a and 13". We checked the other sentences and concluded that they make sense with this revised numbering. The only change we made here is that we added a comma before "which suggests that the gamma distribution is a valid approximation".

Please check if there are other sentences in similar conditions that I did not notice within all the manuscript.

We checked the entire manuscript, but could not find other instances that required attention.

Below some few suggestion:

- Page 8 last paragraph: "....(as shown in Fig??)...."

See response to referee #1. That should have been Fig. 4b. This is now corrected.

- Page 10, line n. 5: please quantify "significantly". Which is the allowed differences between retrieved and TS96 parameters to consider that the gamma distribution is not a good fit?

We employ the qualification "significantly" loosely in this context. We refer to the correspondence (or lack thereof) between the estimated and the 'true' parameters upon visual inspection rather than as a result of

sound statistical testing. To avoid confusion, we replaced the first instance of "significantly" with "appreciably" (1.5) and the second instance with "considerably" (1.6).

- Page 10, line 6: please quantify "significantly"

See previous response.

- Page 10, lines 7-8: Which is the number of time that the measured DSD deviate significantly from the TS96 DSD? Which is the number of time that the estimated DSD deviate significantly from the measured DSD? Please add this information

Note that this is still the Methods section (3), where we do not want to present actual results. The requested information is provided in the Results sections (4 and 5), in particular in Figs. 7 and 8 (Section 4) and Fig. 9 (Section 5) and the corresponding text.

- Table 2 (and all the tables that report the same statistics) : For sake of simplicity please add the unit. fail is a percentage?

All tables report normalized (and hence dimensionless) statistics, as is clearly indicated in the captions. The failure ratio is a (dimensionless) fraction (multiply with 100 to obtain a percentage). This has now also been explicitly stated in Section 3.4.

**General comment:**

The manuscript concerns a retrieval of drop size distribution (DSD) from attenuation of microwave links. Its focus is primarily on numerical validation of a DSD retrieval concept. This rose major concerns of all three reviewers to the initial submission as it was unclear to which extent were the proposed methods applicable for real microwave link observations affected by different sources of errors. The authors took an effort and substantially revised the manuscript. They provided additional numerical analyses investigating effect of signal quantization (precision) on the DSD retrieval and also changed several figures (time series plots to scatter plots) to better illustrate the results.

We thank the referee for his/her positive evaluation of our revised paper as well as for the suggestions for further improvement. We have taken all of them into account in the revised version of our paper (see below for our detailed responses).

The additional analyses show that accuracy of microwave link observations in real networks will greatly affect reliability of the results and substantially limit the feasibility of a DSD retrieval in practice. The authors discuss the feasibility issue in the Discussion section, however, the Conclusions and Abstract section lacks clear and unambiguous statements about feasibility. E.g. the Conclusion section starts with the statement: "Using both simulations and actual link data we have shown that a DSD retrieval on the basis of multiple microwave link variables can be successful and highly accurate, but only when precise high-resolution records of received power are available."

See our response to referee #1 and our response to this referee's next remark. Following his/her suggestion (C4), we have removed "highly" and replaced "received power" by "rain-induced attenuation" in this sentence.

Yes, the authors showed, that a DSD retrieval can be highly accurate on ideal data, but definitely did not demonstrate this high accuracy on real data. And of course, if precise high resolution records would be available, the method would be highly accurate also on real data, but this is not the case. I think that partly negative results are not a reason preventing the manuscript to be accepted for a publication, however, the authors should be much more careful not to give an impression that retrieving accurate DSD estimates from real microwave links is highly feasible. This concerns especially the Conclusion section and Abstract section.

We thank the referee for his/her constructive feedback. We agree that any misconception should be avoided. Following the recommendations of the editor and of referee #1, our assumption of idealized conditions has now been articulated even better throughout our paper, although we would like to recall that we already discussed the practical limitations of our method in the previous version of our paper (e.g. in the last sentence of the abstract and quite extensively in the discussion section). To avoid any misconception on the part of the reader, we have changed the title of our paper to "Estimating raindrop size distributions using microwave link measurements: potential and limitations". In addition, we have replaced the statement "Using both simulations and actual link data we have shown that a DSD retrieval on the basis of multiple microwave link variables can be successful and highly accurate, but only when precise high-resolution records of received power are available" with "Using simulated link data we have shown that a DSD retrieval on the basis of multiple microwave link variables can be successful and highly accurate, but only when accurate, but only when precise high-resolution records of rain-induced attenuation are available. This was confirmed when applied on actual link data, where baseline variations prohibited accurate DSD retrievals".

Furthermore, the manuscript would clearly benefit form more specific conclusions with respect to limitations of the method in practice. For example, the figure 5 together with the table 4 indicates pretty clearly what is the minimal required accuracy of observed attenuation and the range at which it is feasible to estimate the 'mu' parameter. This might be relatively easily linked with minimal rainfall intensity and minimal microwave link length (which affects the sensitivity to rainfall) required for a DSD retrieval.

This is discussed to some extent in the last paragraph of Section 5.4. In particular, we state the following: "To achieve non-convergence ratios of less than 10% a quantization of 0.001 dB or less is required, which makes this not achievable with current generation operational networks. It should also be noted that taking into account the quantization error in the analysis favors the dual-frequency method over the dualpolarization method. This can be attributed to the steeper slope of the attenuation-ratio- $\mu$  relationship within the band of common DSD shapes as shown in Fig. 5". A more specific statement regarding the "minimal rainfall intensity and minimal microwave link length [...] required for a DSD retrieval" can unfortunately not be provided, because this does not only depend on the instrumental characteristics of the microwave link at hand (the length of which in our case was a fixed 2.2 km), but also on the spacetime properties of rainfall (in particular of the DSD). This is definitely something we would like to study in further detail in future work.

Finally, it is difficult to interpret if the obtained results are in fact good or bad. The two-parameter method relies on calibrating relation between parameters of gamma distribution Lambda and mu using disdrometer data (or DSD typical for a local climate), i.e. typical values of Lambda and mu have to be known. It is therefore not clear, what is the real information gain, when using attenuation data compared to the baseline scenario when average (or most common) DSD pdf is assumed to be same for all the events. The results for such baseline scenario (or similar) should be, therefore, included into evaluation. The comparison to 'baseline scenario' might also explain, why three-parameter method (which works independently on calibrated relation between 'Lambda' and 'mu') perform substantially worse than two-parameter method.

The gamma parameterization for the raindrop size distribution (DSD) contains three free parameters ( $N_{T}$ ,  $\mu$  and  $\Lambda$ ). Hence, in principle three independent measurements (in this case with microwave links) are needed to retrieve each of the parameters. However, decades of research concerning the parameterization of DSDs has taught us that the three gamma parameters are correlated to each other, suggesting that the effective number of free parameters is actually less than three. This is also the scientific basis for radar remote sensing of rainfall, both using conventional single-parameter radars (which use measurements of radar reflectivity Z to estimate rain rate R) and using dual-parameter radars (which typically use observations at orthogonal polarizations to estimate R and DSD parameters). The use of a climatological  $\mu$ - $\Lambda$  relationship is a cornerstone of many rainfall retrieval algorithms for dual-parameter weather radar (see e.g. the papers by Zhang et al., 2001, 2003, cited in our paper). Here, we employ exactly the same approach in our dual-parameter rainfall retrieval algorithm, but now applied to microwave links rather than weather radars. In fact, the climatological  $\mu$ - $\Lambda$  relationship we derived from nine months' worth of DSD data from The Netherlands (Eq. (17)) is quite similar to the one proposed by Zhang et al. (2003). Hence, the proposed approach has firm footing in the (radar meteorological) scientific literature. The main reason why the three-parameter method has trouble beating the two-parameter method is probably related to the fact that, besides numerical convergence issues, the effective number of free DSD parameters is rather two than three (due to the mentioned correlations between the parameters). With three measurements the estimation problem then becomes overdetermined. We feel this is discussed in sufficient detail in Section 5.1 of our paper. With respect to the question about the information content compared to assuming a climatological DSD shape, it is clear from the literature that the two parameters of a gamma distribution with  $\mu$  correlated to  $\Lambda$  are nearly orthogonal, and both provide a significant amount of information. Hence, as stated above, having two parameters does add a significant amount of information as compared to just having one parameter (which would be the case if a climatological DSD shape would be assumed).

I recommend the manuscript for a publication after minor revisions concerning mostly presentation and interpretation of the results.

Again, we thank this referee for his/her constructive feedback.

Specific comments:

Fig. 7, 8, and 9: Cannot see orange dots. Maybe transparency of points would help. In addition, consider showing correlation coefficients for all three methods.

The orange dots are indeed invisible in Fig. 7d, can hardly been seen in Fig. 8d and 9d, and clearly fall on the 1:1 line in Fig. 15d. This is a consequence of the fact that the method of moments (of which TS96 is a special case) works very well for the estimation of the rain rate R. Therefore, the use of transparent points would not help. In addition, we do not see the need to show correlation coefficients, as the statistics corresponding to Fig. 7d and 8d are shown in Table 1, those related to Fig. 9d in Table 2, and those related to Fig. 15d in Table 4.

P18L1-3: Is it so, that tree-parameter method can work also without disdrometer data used for calibrating relation between gamma distribution parameters Lambda and mu? If yes, this might be worth to note, because it is important benefit of this method compared to two-parameter method.

This is correct, as has already been noted in Section 3.3 ("In order to still solve for the two parameters an additional equation is required for the relationship between  $\mu$  and  $\Lambda$ "). Note, however, that (as noted in Section 5.1) "the addition of a third microwave link variable does not improve the retrieval (in many cases it actually harms the retrieval) and is unnecessary". Also see our response to one of the previous remarks.

P22L5-6: Consider noting that resulting DSD is not shown.

As suggested by referee #2, we added "(not shown)" to the end of the sentence "The resulting DSD is very similar in shape to that obtained in the simulations, with overestimations especially at smaller diameters, but with the general shape of the DSD preserved".

P26L23-28: The evaluation of baseline scenario with most common Lambda and mu parameters used for all the events would enable more robust reasoning clearly separating effect of DSD observations from the information gain obtained from microwave link attenuation.

As noted on I.25–26, "the  $\mu$ - $\Lambda$  relationship [is] determined from the total of all 9 months of disdrometer measurements, not from the specific event in question". Hence, this is a fixed climatological relationship employed for the entire nine-month period. As such, our method does already separate the "effect of DSD observations from the information gain obtained from microwave link attenuation", as requested by the referee.

P27L11-13: The phrasing gives misleading impression that highly accurate retrieval of DSD was demonstrated also on real data, which is not correct.

We agree. Hence, we decided to replace "both simulations and actual link data" with "simulated link data". Also see our response to referee #1 and our response to one of this referee's previous remarks. Following the suggestion (C4) of referee #1, we have replaced "received power" by "rain-induced attenuation" in this sentence. In summary, as noted in our response to referee #3, we have replaced the statement "Using both simulations and actual link data we have shown that a DSD retrieval on the basis of multiple microwave link variables can be successful and highly accurate, but only when precise high-resolution records of received power are available" with "Using simulated link data we have shown that a DSD retrieval on the basis of multiple microwave link variables can be successful and highly accurate, but only when precise high-resolution records of rain-induced attenuation are available. This was confirmed when applied on actual link data, where baseline variations prohibited accurate DSD retrievals".

Accepted as is.

We thank the referee for his/her positive evaluation of our revised paper.